# Twisted moiré conductive thermal metasurface

Huagen Li [1,8], Dong Wang[2,3,8], Guoqiang Xu[1], Kaipeng Liu[1], Tan Zhang [1], Jiaxin Li[1], Guangming Tao [4], Shuihua Yang [1], Yanghua Lu[5], Run Hu [6], Shisheng Lin [2,7], Ying Li [2,3] ✉ & Cheng-Wei Qiu [1] ✉

Extensive investigations on the moiré magic angle in twisted bilayer graphene have unlocked the emerging field—twistronics. Recently, its optics analogue, namely opto-twistronics, further expands the potential universal applicability of twistronics. However, since heat diffusion neither possesses the dispersion like photons nor carries the band structure as electrons, the real magic angle in electrons or photons is ill-defined for heat diffusion, making it elusive to understand or design any thermal analogue of magic angle. Here, we introduce and experimentally validate the *twisted thermotics* in a twisted diffusion system by judiciously tailoring thermal coupling, in which twisting an analog thermal magic angle would result in the function switching from cloaking to concentration. Our work provides insights for the tunable heat diffusion control, and opens up an unexpected branch for twistronics -- twisted thermotics, paving the way towards field manipulation in twisted configurations including but not limited to fluids.

The emergence of various metamaterials[1–7] has brought considerable interesting and counterintuitive phenomena to us humans such as invisibility cloak over the decades. Thereby, it has attracted extensive interest from many researchers and various metamaterials have been successively reported in numerous fields, e.g., electromagnetic, thermodynamics[8–15], etc. Among them, various thermal metamaterials or metadevices[16–20] have been proposed and designed to realize the vision of controlling heat flow at will. Specifically, they can achieve unusual thermal properties or functions such as thermal transparency[13], thermal concentration[21], and thermal rotation[16] beyond natural materials based on the transformation theory[11,22] or scattering cancellation technique[17,23]. Recently, various schemes of actively or passively tuning thermal conductivities have been proposed and designed, including the switchable[21,24], doublet[25], tunable solid-like convective[26], near-zero-index[27], tunable analog[28], and passive ultrathin schemes[29]. However, these schemes are all restricted to either the background material or the in-plane modulation. Therefore, another important tool without these above restrictions—twisted thermotics is urgently needed.

Recently, the emerging field—twistronics[30] has drawn enormous interest, especially the twisted bilayer graphene which can generate intrinsic unconventional superconductivity[31] or counterintuitive insulator behaviors[32] at a certain small magic angle that induces a flat band in strongly correlated systems, originating from the transformation of electrical conductivity tensor as a function of the angle. In photonics, analogous concepts have been reported in

[1]Department of Electrical and Computer Engineering, National University of Singapore, Kent Ridge 117583, Republic of Singapore. [2]State Key Laboratory of Extreme Photonics and Instrumentation, ZJU-Hangzhou Global Scientific and Technological Innovation Center, Zhejiang University, Hangzhou 310027, China. [3]International Joint Innovation Center, Key Lab. of Advanced Micro/Nano Electronic Devices & Smart Systems of Zhejiang, The Electromagnetics Academy of Zhejiang University, Zhejiang University, Haining 314400, China. [4]Wuhan National Laboratory for Optoelectronics and State Key Laboratory of Material Processing and Die and Mould Technology, School of Materials Science and Engineering, Huazhong University of Science and Technology, Wuhan 430074, China. [5]Smart Materials for Architecture Research Lab, Innovation Center of Yangtze River Delta, Zhejiang University, Jiaxing 314100, China. [6]State Key Laboratory of Coal Combustion, School of Energy and Power Engineering, Huazhong University of Science and Technology, Wuhan 430074, China. [7]Chongqing 2D Materials Institute, Chongqing 400015, China. [8]These authors contributed equally: Huagen Li, Dong Wang. ✉e-mail: eleying@zju.edu.cn; chengwei.qiu@nus.edu.sg

succession via these thin photonic crystals in twisted two-dimensional (2D) materials[33–38]. Undoubtedly, these extraordinary electronic or photonic features related to moiré superlattices have greatly expanded the developing field of twistronics[30]. However, the concept of the magic angle possesses far-reaching implications beyond its original domain, holding great potential for diverse scientific applications. Beyond the traditional confines of magic angle in electronics and photonics, the magic angle concept has the potential to permeate various scientific disciplines, pushing the boundaries of knowledge and innovation. For instance, recent studies have explored the intriguing connection between magic angle phenomena and quantum information science, highlighting the possibility of using twisted systems as platforms for quantum computing and simulation[39]. Moreover, the magic angle concept has also shown promise in fields such as biophysics, where the manipulation of twist angles in DNA molecules has revealed unexpected structural and functional properties[40,41]. Notably, no matter either in electronics or photonics, there are always numerous counterparts in conductive heat transfer. Therefore, inspired by the general characteristics among diffusion, electronics, and photonics, an analog twisted bilayer thermal metamaterial to actively tune conductivity tensor as its thermal counterpart can be proposed and designed, realizing a similar magic angle for heat diffusion. Besides, despite extensive theoretical and experimental efforts to explore the precise control of heat diffusion, unconventional heat conduction in a twisted bilayer system is challenging and nontrivial to study theoretically because of the inadaptability of the typically used models, motivating the potential theories and feasible methods for investigating these diffusion systems. Nevertheless, the specific design principle in a twisted diffusion system needs to be further explored.

Specifically, for a conventional bilayer thermal cloak, there are two concentric shells with two different radii. It can only realize the cloak effect[27] due to the fixed thermal conductivity. We may need to renew the components or change the structure for the realization of a switchable thermal metadevice. Notably, the thermal conductivity in an anisotropic bilayer structure such as a bilayer stripe structure can be tuned via twisting different angles. Nevertheless, these corresponding theoretical models are challenging to investigate because of the invalidation of the traditionally used models. Meanwhile, since heat diffusion neither possesses the dispersion like photons nor carries the band structure as electrons, the concept of magic angles in twistronics or opto-twistronics is absent for heat diffusion, resulting in the difficulty of designing any thermal magic angle. Admittedly, given the similarity between electrical and thermal conductivity tensors, the "twisted thermotics" where the coupling emerges within a twisted diffusion system can be thus introduced.

In this work, inspired by the magic angles in twistronics or opto-twistronics, we propose a twisted thermal metadevice for the proof-of-concept, possessing a potential advantage for the rapid deployment of metadevices with various functions. It introduces the "twisted thermotics" for heat diffusion. Based on effective medium theory, the effective thermal conductivity tensor of the twisted stripe region is derived as a function of the angle. Moreover, the width $w$ of the stripes and radius $R_1$ theoretically elicits the so-called thermal magic angle, endowing a similar inflection point owing to the alternate permutations of stripes. Then, we numerically and experimentally demonstrate the analog thermal magic angle via twisting the line moiré patterns, exhibiting a switchable effect from cloak to concentration. The thermal analog of magic angles in twistronics or opto-twistronics is thus discovered. Besides, the rotation and thermal Janus effect via twisted manipulations are also shown in simulations. In contrast to previous approaches, our twisted thermal metadevice provides more inspirations for the tunable conductive heat manipulation and significantly opens an avenue for the twistronics−twisted thermotics.

## Results

### Origin of thermal magic angle and twisted thermotics

As we know, the twisted bilayer graphene with moiré pattern can generate unconventional superconductivity or counterintuitive insulator behaviors by twisting a small magic angle[31,32]. Then, the topological polaritons and photonic magic angles as optics analogs have been discovered in succession via twisting α-MoO₃ bilayers[35]. Similarly, twisted stripe bilayers as shown in Fig. 1a, in principle, can introduce extreme anisotropy, indicating the possibility to realize a dramatic change of diffusion via twisting a tiny angle. Besides, in terms of the microscopic scale, the magic angle in thermal conductivity of twisted bilayer graphene has been theoretically discovered[42]. In contrast to these traditional models, twisted stripe bilayers in this proposed system can provide another degree of freedom to control heat flow at will, leading to more counterintuitive phenomena. The potential mechanism might lie in the thermal analog of magic angles in twistronics or opto-twistronics. In electrons or photons, conductivity tensor can be heavily influenced by tuning to these magic angles that induce a near-zero Fermi surface. Likewise, effective thermal conductivity tensor also can be tuned by a twisted bilayer diffusive system as shown in Fig. 1b and d (the original angles of the upper and lower layers compared with the positive direction of $x$-axis are set as $\alpha_1 = \alpha_2 = 90°$ and the direction of heat flux about the positive direction of $x$-axis is $\varphi = 0$). Meanwhile, the measurement of effective thermal conductivity is acquired by a minimum mean-square error (see Supplementary Note 1). Herein, our primary aim is to macroscopically modulate heat exchange coupling based on the twisting angle. As previously established, twisted bilayer graphene has the ability to influence electrical conductivity as a function of the angle, showcasing a magic angle phenomenon for electrons[31,32].

We observed an inflection point in thermal conductivity by twisting our proposed stripe bilayer system, as an analog to the previously reported graphene's magic angle[31,32]. Notably, the significant alteration in the effective thermal conductivity tensor or temperature gradient, referred to as the formation of the "thermal magic angle", can be approximately determined based on the structural parameters of these stripes (as depicted in Fig. 1c).

The implications of this analog thermal magic angle in our system are vast. It opens avenues for various dynamic thermal management applications, including but not limited to thermal cloak, thermal concentration, or thermal rotation. Essentially, the thermal analog of the magic angle introduced here can lead to the realization of a multifunctional switching effect (as illustrated in Fig. 1e). This comprehensive exploration and understanding of the analog thermal magic angle paves the way for diverse applications and advancements in related fields.

### Twisted thermotics theory

For heat diffusion, we are mainly focused on the steady-state temperature field $T$ that obeys Fourier's law[18] as

$$\kappa \nabla^2 T = 0 \tag{1}$$

Therefore, for the steady-state temperature fields, the potential heat exchange coupling mechanism might lie in the change of thermal conductivity tensor $\kappa'$ with respect to the original thermal conductivity tensor $\kappa$ after the coordinate transformation. Now considering each stripe layer (upper or lower) with different thermal conductivities $\kappa_I$ and $\kappa_{II}$, each composed of two kinds of stripe widths $w_I$ and $w_{II}$, the original anisotropic thermal conductivity tensors $\kappa_{upper}^{eff}$ and $\kappa_{lower}^{eff}$ for both layers can be easily acquired (see Supplementary Note 2). Then, assuming that the original angles of the upper and lower layers compared with the positive direction of $y$-axis are both set as $\alpha$ and the direction of heat flux about the positive direction of $x$-axis is $\varphi = 0$, the modified thermal conductivity tensor based on its form invariance

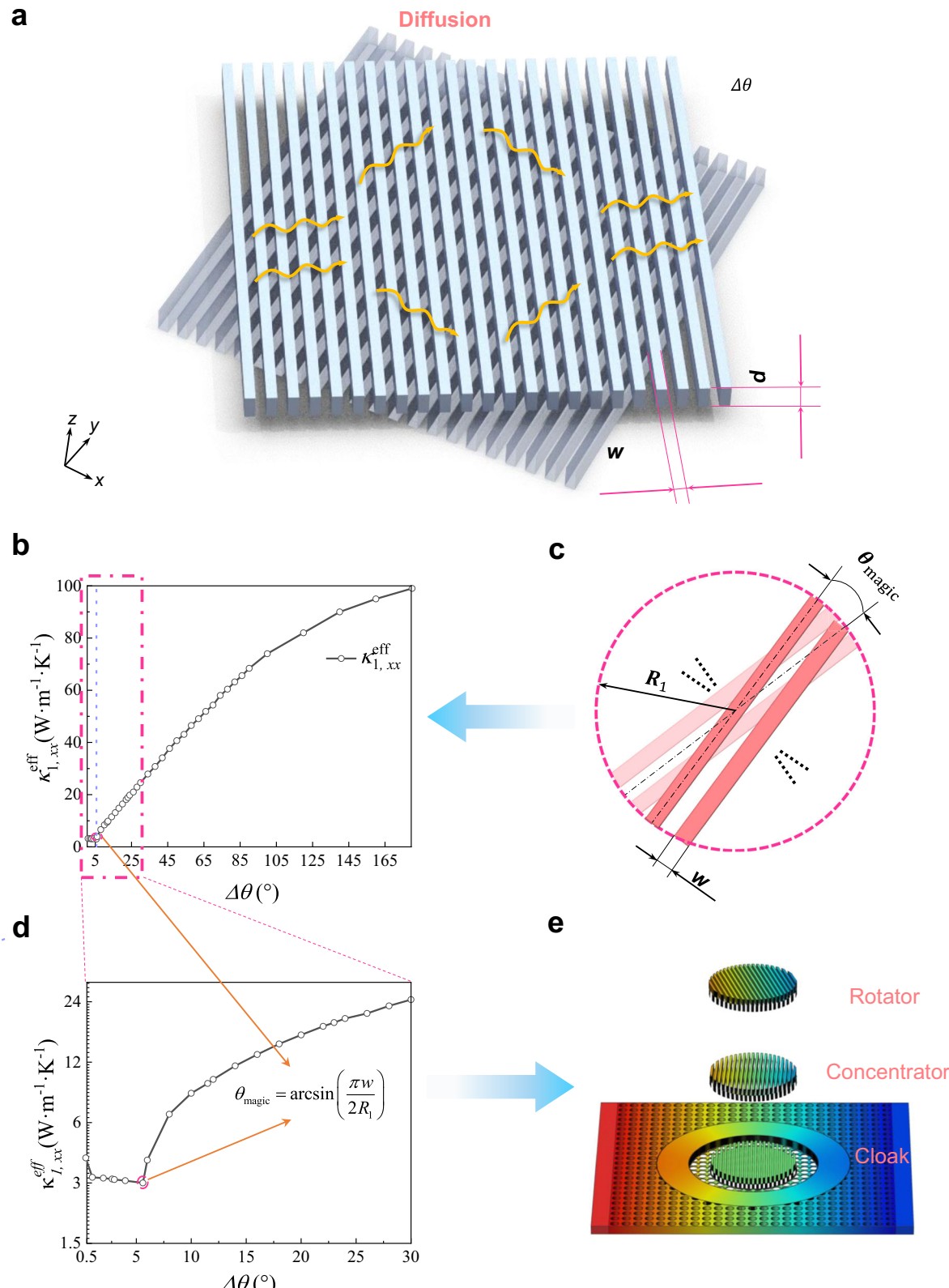

**Fig. 1 | Origin of twisted thermotics. a** The diffusion systems consisting of twisted stripe bilayers within a Cartesian coordinate framework. **b** The curve of effective thermal conductivity $\kappa_{1,xx}^{\text{eff}}$ measured by a minimum mean-square error. **c** The analog thermal magic angle $\theta_{\text{magic}}$ related to the structure parameters $w$ and $R_1$. **d** The enlarged curve of effective thermal conductivity $\kappa_{1,xx}^{\text{eff}}$ around the analog thermal magic angle $\theta_{\text{magic}}$. **e** Schematic diagram of the multifunctional switching effect related to the analog thermal magic angle. The thickness, width, and radius of the upper and lower stripe layers are denoted by $d$, $w$, and $R_1$, respectively.

under coordinate transformation is shown as (see Supplementary Note 2 in detail)

$$\kappa_{\text{upper}}^{\text{eff}}\Big|^{\alpha} = \frac{J_1 \kappa_{\text{upper}}^{\text{eff}} J_1^T}{\det(J_1)} \tag{2}$$

where $J_1$ is the Jacobian matrix of the coordinate transformation between the original versus modified coordinate systems, $J_1^T$ and $\det(J_1)$ are the transpose and the determinant of the $J_1$, respectively. Similarly, we can obtain the $\kappa_{\text{lower}}^{\text{eff}}\Big|^{\alpha}$ of lower layer which is equal to that of upper layer due to the two layers with the same original angle $\alpha$.

According to the effective medium theory[43], effective thermal conductivity $\kappa^{\text{eff}}$ can be achieved after twisting the upper layer an angle $\theta_1$ and the lower layer an angle $\theta_2$, and we can obtain an effectively anisotropic thermal conductivity tensor which is different from that proposed in one previous work[27]. The expression of the corresponding thermal conductivity tensor in the Cartesian coordinate is written as (see Supplementary Note 2 in detail)

$$\kappa^{\text{eff}} = \begin{pmatrix} \kappa_{xx}^{\text{eff}} & \kappa_{xy}^{\text{eff}} \\ \kappa_{yx}^{\text{eff}} & \kappa_{yy}^{\text{eff}} \end{pmatrix} \tag{3}$$

## Theoretical reconfiguration of thermal magic angle

Currently, based on the above-twisted thermotics theory, we can twist the two-line moiré patterns to modulate the effective thermal conductivity tensor, resulting in judiciously tailoring the thermal coupling as a function of the angle. The general form of thermal conductivity tensor is theoretically described as the above Eq. (3). However, in contrast to the magic angle phenomena of the electrical conductivity, the general form of thermal conductivity tensor seems to be trivial due to the lack of some counterintuitive and interesting phenomena such as a similar magic angle phenomenon in a designed thermal conductivity tensor.

Interestingly, if the stripes with the same thermal conductivity are alternately arranged on two layers and the width of the stripes should meet the required condition ($w_{\text{I}} \geq w_{\text{II}}$), we can acquire a counterintuitive and different thermal conductivity $\kappa_{xx}^{\text{eff}}$ along the $x$-axis, which is larger than that of the series conductivity at the original state with zero degrees based on the related literatures[25,44,45]. Moreover, given the practical applications of the twisted diffusion system, the zone with stripes cannot be infinitely large so that we can acquire a similar inflection point of thermal conductivity tensor at a small angle (Fig. S2) due to the influence of the structure parameters. As we know, when the direction of heat flux is along the positive direction of $x$-axis, it is easy to find out that twisting the stripes to form a zigzag structure can maximumly reduce the heat diffusion (Fig. S3). Thus, the current form of theoretical reconfiguration of the thermal magic angle is described as (see Supplementary Note 3 for the other case in a circular region)

$$\theta_{\text{magic}}\Big|_{\text{rec}} \approx \arcsin\left(\frac{w_{\text{I}} + w_{\text{II}}}{L}\right) \tag{4}$$

where $L$ is the length of two different stripes in a rectangular region.

## Extended theoretical principle in twisted metadevices

Now, considering there is just a twisted bilayer system shown in Fig. S6, we can easily obtain the general solution and matching function of this system (see Supplementary Note 4). For simplicity, the upper and lower stripe layers in the circular region IV are composed of the evenly spaced stripes with width $w$ and different thermal conductivities $\kappa_{\text{I}}$ and $\kappa_{\text{II}}$. Then, according to the above-twisted thermotics theory, the

effective thermal conductivity tensor $\kappa_1^{\text{eff}}$ of the region IV in twisted metadevices can be achieved through the heat exchange coupling at the interface of the two stripe layers after twisting the upper layer an angle $\theta_1$ and the lower layer an angle $\theta_2$ (see Supplementary Note 4). Additionally, the heat flux bending angle $\phi_1(\theta_1,\theta_2,\alpha)$ can be expressed as

$$\phi_1(\theta_1,\theta_2,\alpha) = \tan^{-1}\left(\frac{\kappa_{1,yx}^{\text{eff}}}{\kappa_{1,xx}^{\text{eff}}}\right)$$
$$= \tan^{-1}\left(\frac{(-\kappa_x + \kappa_y)[\cos(\theta_1 - \alpha)\sin(\theta_1 - \alpha) + \cos(\theta_2 - \alpha)\sin(\theta_2 - \alpha)]}{\kappa_x[\cos^2(\theta_1 - \alpha) + \cos^2(\theta_2 - \alpha)] + \kappa_y[\sin^2(\theta_1 - \alpha) + \sin^2(\theta_2 - \alpha)]}\right) \tag{5}$$

More importantly, compared with the phenomena in the decoupled stripe bilayer (Fig. S7b–d), the heat exchange coupling strength (Fig. S7a) becomes weaker and weaker before reaching the thermal magic angle induced by structural parameters, and it only keeps the coupling mainly originating from the off-diagonal items of effective thermal conductivity tensor $\kappa_1^{\text{eff}}$ after surpassing the thermal magic angle via the symmetric twisted manipulation. Considering the influence of the stripe width $w$ in a circular region as shown in Fig. S4, the corresponding magic angle $\theta_{\text{magic}}$ in this twisted metadevice can be derived as

$$\theta_{\text{magic}} = \arcsin\left(\frac{\pi w}{2R_1}\right) \tag{6}$$

## Theoretical verification of thermal magic angle in twisted metadevices

To verify our twisted model designed above, the finite-element simulations have been performed with the software COMSOL MULTIPHYSICS. Firstly, for the background with arbitrary thermal conductivity, the corresponding thermal conductivity $\kappa_3$ of region II can be derived based on the corresponding matching function. Then, this metadevice still works with the change of $\kappa_1^{\text{eff}}$ determined by Eq. (S19) via twisted manipulations owing to the satisfaction of the matching function.

To start, the temperature profiles of the theoretical twisted model are presented in Fig. 2a, b. Obviously, it is easy to understand that the heat flux bending angle disappears by twisting an equal angle $\theta_1$ and $\theta_2$ in opposite directions based on the heat flux direction $\varphi$, respectively. It can be observed from the simulation results (Fig. 2a) that when the twist angles are both zero degrees, the isotherms of the background are almost without distortion and those in region IV almost vanish, exhibiting the realization of a thermal cloak. However, since the effective thermal conductivity in Region IV changes dramatically with a tiny increase of twisted angles, the isotherms of the background are almost still without distortion while the interval of the isotherms in Region IV turns out to be narrower compared with those of the background, indicating the realization of a cloak-to-concentrator. Moreover, for the central region, the effective thermal conductivity and temperature gradient along the measured line of $y = 0$ cm with corresponding simulated results as a function of twisted angle difference ($\Delta\theta = \theta_1 - \theta_2$) are presented in Fig. 2c, d, respectively. Interestingly, it is easy to observe that the positions of peak value of numerical curves (Fig. 2d) with the decrease of the stripe width $w$ are governed by Eq. (6). Meanwhile, the width $w$ of stripe and radius $R_1$ could theoretically elicit the maximum value of temperature gradient related to the twisted angle, indicating the possibility to imitate a thermal magic angle. In short, when the thermal analog of the magic angle governed by Eq. (6) is achieved and the effective thermal conductivity in region IV can be approximately calculated by Eq. (S19), the switchable effect emerges.

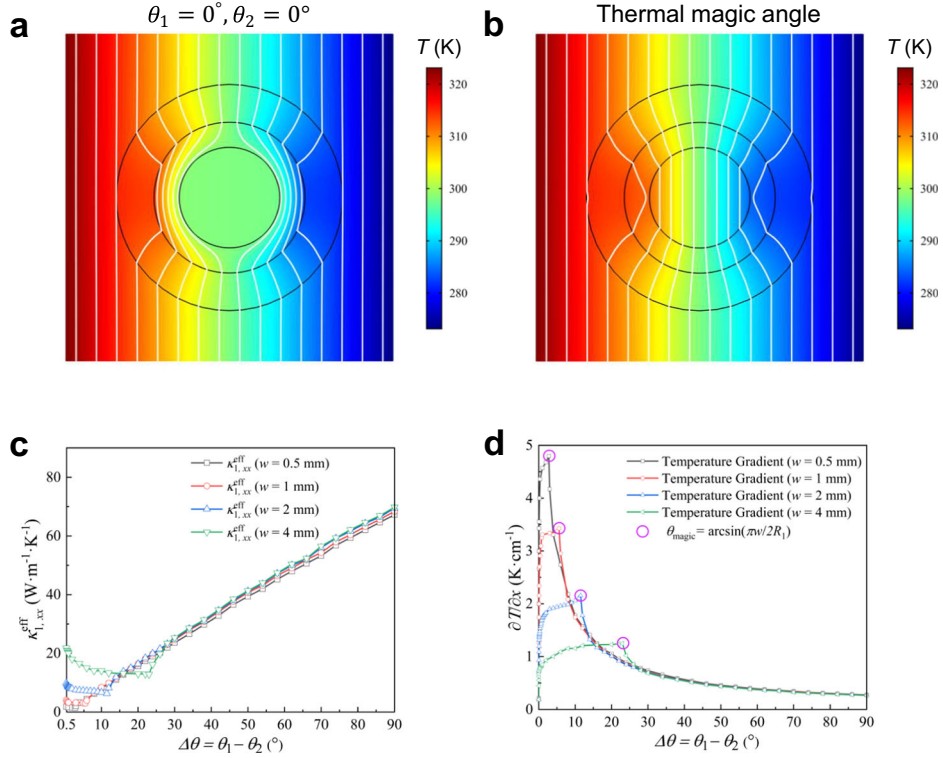

**Fig. 2 | Theoretical and simulated twisted thermotics. a**, **b** Results of the temperature profiles for the twisted angles $\theta_1(\theta_2)$ with **a** $\Delta\theta = 0°$ and **b** $\Delta\theta = \theta_{\text{magic}} = \arcsin(\pi w/2R_1)$. **c**, **d** The dependences of **c** the effective thermal conductivity $\kappa^{\text{eff}}_{1,xx}$ and **d** temperature gradient on the angle difference ($\Delta\theta = \theta_1 - \theta_2$).

## Numerical and experimental demonstration of thermal magic angle phenomenon

To further demonstrate the imitated magic angle, the corresponding simulation model is designed based on the experimental setup as shown in Fig. 3a (see Supplementary Note 5: Fig. S8 for the actual structure), where only aluminum alloy is used to construct the desired structure. Herein, the original angles of the upper and lower layers compared with the positive direction of $x$-axis are set as $\alpha_1 = \alpha_2 = 90°$ and the direction of heat flux about the positive direction of $x$-axis is $\varphi = 0$. The simulated temperature distributions for the two different twisted conditions are presented in Fig. 3b, c. Likewise, the corresponding measured temperature profiles for the two actual twisted situations are depicted in Fig. 3d, e, respectively. Obviously, the temperature distribution of the numerical and experimental results is easily observable, allowing for a straightforward comparison. Therefore, it is evident that the numerical and experimental results align closely, indicating a high degree of similarity. Besides, by examining the numerical and experimental results (Fig. 3b, d), it is evident that when the twist angles are set to zero degrees, the isotherms of the background exhibit minimal distortion, while those in region IV nearly disappear, illustrating the successful realization of a thermal cloak. Similarly, with a slight twist angle of $2.8°(-2.8°)$, the interval between the isothermal lines in region IV is narrower compared to the background, as observed in the numerical and experimental results (Fig. 3c, e). This observation suggests a concentration of thermal effects in that region. For the simulated and actual models, the temperature variations along the measured line of $y = 0$ cm are presented in Fig. 3f, g, respectively. Moreover, the experimental results agree well with the simulated results. Accordingly, the thermal analog of magic angle is thus discovered numerically and experimentally.

## Extended multifunctional effect of twisted thermotics

Then, as the value of $\kappa^{\text{eff}}_1$ is further changed by Eq. (S19), we can easily achieve the thermal Janus effect via twisted manipulations as

illustrated in Fig. 4a, b, respectively. Obviously, for the case in Fig. 4a, when the heat flux is along the positive direction of $x$-axis, the isothermal lines in the central region are constant and the isotherms of the background are without distortion, exhibiting a cloaking effect. Interestingly, when the heat flux comes from the negative direction of the $y$-axis, the interval of isothermal lines in region IV is similar to that in the background (Fig. 4b), indicating an enhanced thermal transparency. Besides, when the twisted angles $\theta_1(\theta_2)$ in region IV are $15°(0°)$, the thermal rotation effect can be observed from Fig. 4c due to the formation of the heat flux bending angle governed by Eq. (5). Interestingly, it is easy to observe that the Janus effect still exists by comparing Fig. 4c, d. Furthermore, when the twisted angles $85°(-85°)$ (Fig. 4a) in region IV are tuned to be $15°(0°)$ (Fig. 4c), the switchable cloak-to-rotation can be achieved. In brief, it indicates that various switchable thermal effects such as cloak-to-transparency, cloak-to-rotation, or transparency-to-rotation can be easily and quickly realized in this twisted metadevice.

## Discussion

In summary, we propose a method to render a twisted diffusive system with the concept of an analog thermal magic angle and establish the potential theory of twisted thermotics. This twisted bilayer thermal metadevice can realize the switchable thermal functions with near-zero energy input, possessing a potential advantage for the rapid deployment of multifunctional thermal metadevice. By considering the interlayer coupling of the effective thermal conductivity tensor and leveraging the benefits of graphene coatings, we can investigate more complex thermal fields and gain greater control over heat flux in material systems. The understanding of interlayer coupling and the use of graphene coatings offer exciting opportunities for enhancing thermal transport and achieving desired heat transfer effects. By pushing the boundaries of our understanding in this field, we contribute to the advancement of thermal management across various applications, resulting

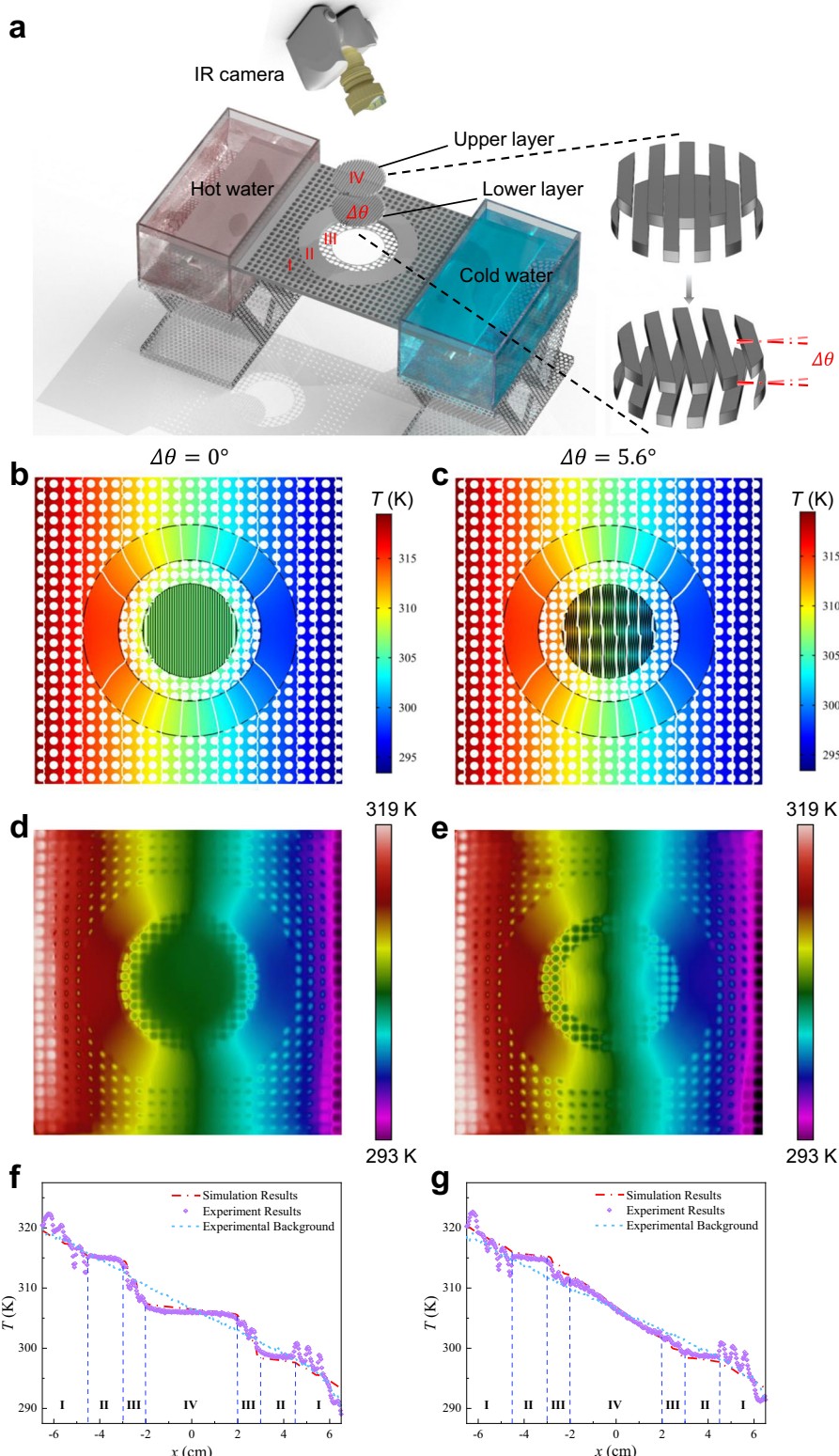

**Fig. 3 | Numerical and experimental demonstration of thermal magic angle.**
**a** Schematics of the entire experimental setup. **b**–**e** Temperature distributions of thermal cloak-to-concentration for two different combinations of twisted angles.

**f**, **g** Temperature distributions along the line $y = 0$. The lines of the background (blue), simulation results (red), and experimental data (purple) are presented, respectively.

in elevated performance and efficiency in heat-related processes. Notably, the width of the stripe and radius $R_1$ could theoretically elicit the switchable effect via a small symmetric twisted angle. It indicates the feasibility to obtain the thermal analog of the magic angle. Furthermore, the thermal magic angle is also numerically and experimentally demonstrated, indicating that it can be robust in various conditions. Besides, other thermal functions such as rotation and the Janus effect could also magically emerge at certain

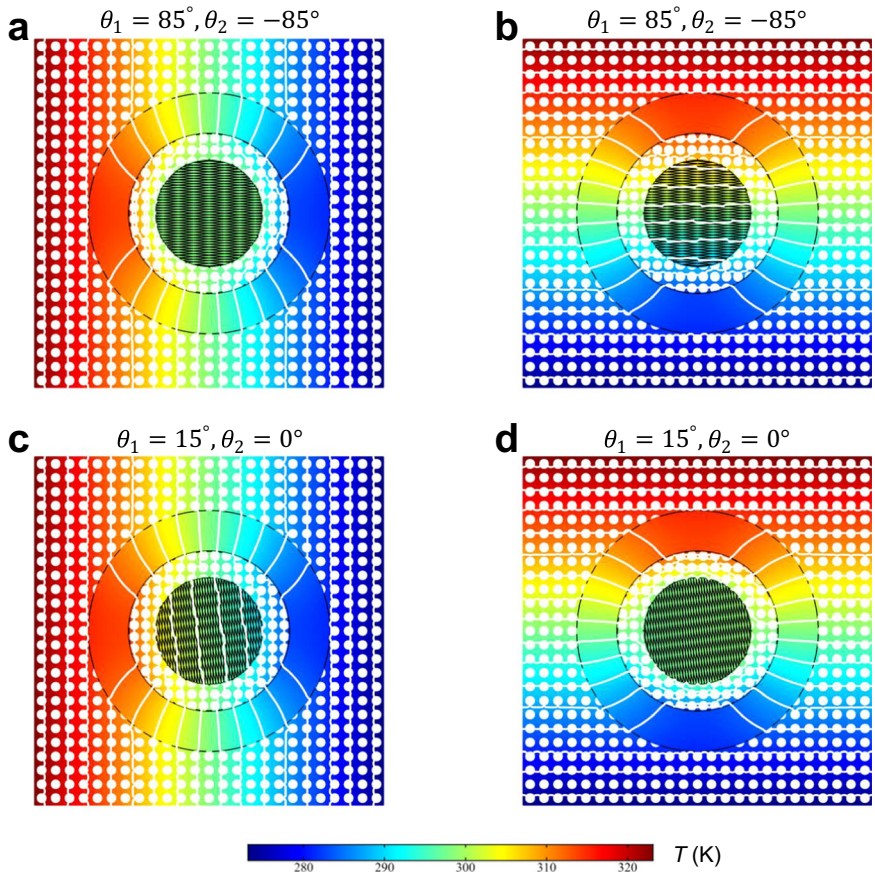

**Fig. 4 | Temperature profiles of thermal metadevice with the thermal Janus effect for two different combinations of twisted angles. a** Thermal cloak. **b** Enhanced thermal transparency. **c** Thermal rotator. **d** Thermal cloak.

twisted angles. In contrast to previous approaches, our twisted bilayer diffusive scheme with near-zero energy input is suitable for multifunctional thermal metadevices with high tunability and flexibility. Our work is expected to open more thermal modulation inspirations and provides an extended theoretical framework for the twistronics−twisted thermotics, paving the way towards other twisted diffusive fields, such as particles and fluids.

## Methods
### Numerical simulations
The whole schemes with different twisted angles are set in a 130 mm × 130 mm square background. The outermost and second outer radii of this structure are 45 mm and 30 mm respectively. Given the matching function shown in Eq. (S30), a circular hole of 2 mm is used with a hole interval of 5.2 mm in the background, and a circular hole of 2.1 mm with a hole interval of 4.3 mm is set in the second outer region. For the central domain, the upper and lower stripe layers with a thickness $d$ and width $w$ of 10 mm and 1 mm are filled to induce the proposed twisted field. For the whole designed schemes, the steady-state numerical results are performed through COMSOL Multiphysics 5.4a. During the related simulations, the different twisted angles $\theta_1$ and $\theta_2$ are employed on the upper and lower layers in the central region, respectively. For the left and right boundaries, the high-temperature line (319.45 K) and low temperature line (293.45 K) are used, respectively. Besides, a twisted trilayer diffusive system has been numerically carried out (see Supplementary Note 6: Fig. S9 for the numerical results) and an elliptic boundary condition has been further investigated to generalize our approach (see Supplementary Note 7). The introduction of graphene coatings in our twisted bilayer diffusive

systems offers an exciting opportunity to expedite the formation of steady-state thermal coupling phenomena, as demonstrated in Fig. S11 (see Supplementary Note 8).

### Experiments
The fabricated metadevices have the same geometry shown in Fig. 3a. The whole structures are made of only aluminum alloy by 3D printing and all the surfaces are coated by graphene via a spraying device. Herein, the type of aluminum alloy and graphene are $AlSi_{10}Mg$ and graphene nanopowder, respectively. The graphene nanopowder was purchased from Graphene Supermarket. Thermal contact resistance[46,47] plays a significant role in the performance of thermal metadevice if this device is constructed by assembled components with some weld joints. Since the lack of weld joints via 3D printing and these used graphene coatings, the effects of thermal contact resistance can be neglected with the current implementations. The temperature profiles of the whole experimental demonstration are acquired by an IR camera (FOTRIC 225 S). The temperatures are $T_1 = 319.45$ K and $T_2 = 293.45$ K. The whole system is covered in a thin polypropylene film with a high emissivity (0.97), leading to an accurate measurement of the temperature field by the IR camera.

## Data availability
All technical details for producing the figures are enclosed in the Supplementary Information. The experimental data generated in this study are provided in the Source Data file. Additional data that support the findings of this study are available from the corresponding author (C.-W.Q. and Y. Li) on request. Source data are provided with this paper.

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

## Acknowledgements

Y. Li acknowledged the support by the Key Research and Development Program of the Ministry of Science and Technology under Grant No. 2023YFB4604100. C.-W.Q. acknowledged the financial support by the Ministry of Education, Republic of Singapore (Grant A-8000107-01-00). Y. Li acknowledged the support by the National Natural Science Foundation of China under Grants Nos. 92163123 and 52250191, and Zhejiang Provincial Natural Science Foundation of China under Grant No. LZ24A050002. G.T. acknowledged the support by the National Natural Science Foundation of China (62175082).

## Author contributions

C.-W.Q. conceived of the idea. H.L., Y. Li, and C.-W.Q. designed the research. H.L. performed the theoretical derivations and numerical simulations. H.L., D.W., Y. Li, and C.-W.Q. designed and conducted the experiments. Y. Lu and S.L. fabricated the graphene coatings. H.L., D.W., G.X., K.L., T.Z., J.L., G.T., S.Y., R.H., Y. Li, and C.-W.Q. analyzed the data. H.L., G.X., J.L., S.Y., Y. Li, and C.-W.Q. made the visualization and wrote the manuscript. All the authors contributed to the discussion and manuscript editing. Y. Li and C.-W.Q. supervised the work.

## Competing interests

The authors declare no competing interests.
