## [Peer Review File · Nature Communications]

REVIEWER COMMENTS

Reviewer #1 (Remarks to the Author):

The authors describe theoretical and simulation work on a structure with two thermally conductive layers twisted by an angle relative to each other. The authors demonstrate some interesting thermal functionalities that emerge out of this twisted system. However, the work is done entirely in the diffusive regime, assuming the heat diffusion equation and Fourier law, as implemented in the commercial tool COMSOL, are valid. The dimensions of the stripes in the two layers are not specified so it is impossible to say whether they are macroscopic and HDE is valid or if they are microscopic and HDE is not valid. The experimental work seems to indicate the former. However, the experimental work does not appear to be on striped layer but rather a circular cloaking setup with equally spaced holes in the material.

The other large issue with this manuscript for me is that the authors use the analogy to twisted bilayer graphene throughout and there really is no connection whatsoever. TBG creates flat bands microscopically through interlayer interactions and the Moire pattern. The flatness of the bands is crucial and emerges from the twist angle. Here the authors do not explicitly consider any coupling between the two layers as a function of the angle. Further, they state that "heat diffusion neither possesses the dispersion like photons nor carries the band structure as electrons, the concept of magic angles in twistrionics or opto-twistrionics is absent for conductive heat transfer". This is strictly only true in the diffusive regime. Heat transfer can also be hydrodynamic, which has been demonstrated in graphene, and then it has a dispersion that can be impacted by twist. Further, at the microscopic scale, phonons possess a dispersion and it has been demonstrated to be impacted by twist. Therefore, the limitation arises out of the assumption that heat is purely diffusive, which is only valid at macroscopic scales. Thus this work has very little to do with twisted bilayer materials.

Reviewer #2 (Remarks to the Author):

The authors report an interesting work related to twisted material systems. The manuscript discusses the existence of a thermal analogue of the magic angle, identified with the term twisted thermotics in a twisted diffusive system. The proposed methodology is of interest for the readership of twisted-systems and materials containing moiré-superlattices. The method presents the concept of imitated thermal magic-angle. Can the authors better explain this concept in the manuscript? In Fig.1A the twisted stripe system is not clear. Does it refer to a twisted-graphene stripe or to a more general system. It would be helpful for the reader to see a direct applicability of the method. When considering twisted 2D-systems with certain chemical composition, would this method be still applicable? Would the stripe-fabrication affect the overall crystallinity of samples? In twisted graphene, van Hove singularities can be significantly altered or weakened by the the twist-disorder parameter or by defects, this makes reproducibility

extremely challenging. Can the authors provide practical examples of as fabricated twisted stripe-systems, with scanning tunneling microscopy or transmission electron microscopy? Would this type of system still exhibit van Hove singularities, can the authors provide an example?

Reviewer #3 (Remarks to the Author):

Motivated by the emerging field of twistronics, this work proposes the concept of “twisted thermotics”, which seems interesting at first glance. However, after carefully going through the manuscript, I think there are quite a few important issues to be clarified or resolved before it can be considered for publication. In particular, there are many writing problems which make the manuscript fairly hard to follow.

Below are some of my major concerns/questions.

1. About the basic idea

- What is twisted thermotics? Is it a device or a field?
- What makes a diffusive system special in the study of twist-angle-induced effects?
- What is the thermal magic angle? What makes it magic?
- Why is there a thermal magic angle in the proposed system? How are the two layers of stripes coupled? How does their coupling vary with the twist angle?
- What is the relation between the thermal magic angle here and the magic angles of twisted bilayer graphene?

2. About the theoretical/numerical study

- Most of the subscripts are not properly explained.
- Many of the symbols and formulas are not easy to follow.
- Some of the derivations in the SI should probably go to the main text.
- The functional regions are not clearly marked in Fig. S2. An appropriate schematic of the device in a main figure may help.

3. About the experimental measurements

- Materials

- o What is “aluminum material”?
- o What kind of graphene is used for the coating?
- o What is the role of the thin polypropylene film?
- o How about the surface roughness?
- How are the fixed-temperature boundary conditions applied?
- What are the uncertainties for measuring the temperature distributions?
- “Herein, the measurement of effective thermal conductivity is acquired by a minimum mean-square error” What does this mean?
- Is there an interfacial thermal resistance between the two layers? Does it matter?

My minor concerns are mostly with the writing. There are many typos, grammatical issues, and inaccurate expressions, some of which are listed below as examples.

- Please try to limit the use of phrases like “quite challenging”, “quite numerous”, “highly nontrivial”, “significantly opens up”, “extremely tuned”. The importance of a work lies in itself, not the adjectives.
- Please pay attention to the use of “the”.
- Please pay attention to the use of “,” before “and”.
- “twisted modulations” and “twisted manipulations” are not grammatically sound.
- “it may waste the precious time”
- “extreme thermal anisotropic”
- “the incomprehensibility of any thermal magic angle”
- “making it be expanded and generalized”
- “via twisting twisted α -MoO₃ bilayers”
- “twisted bilayer stripe”
- “the relative thickness of ultra-thin wall”
- “enlarge the relatively effective thermal conductivity”
- “super high-low structure”
- “synthesize another freedom”
- “upper and under layers”
- “the down layer”

- “imitated thermal magic angle”
- “rendered a similar conception”
- “Now, considering there is just a twisted bilayer system shown in Fig. S2. Then, we can easily obtain the general solution and matching function of this system”
- “can be achieved between the interface of the two layers”
- The expression of kx
- The use of brackets (())
- “for the region III, it is composed of”
- “Especially, for the twisted model”
- “arbitrarily thermal conductivities”
- “realization of a twisted cloak-to-concentrator”
- “which all the structures with only the aluminum material”
- “secondary outer”
- “as shown in below” in the SI

Overall, I am afraid this work does not meet the high standards of Nature Communications, and may be suitable for a more specialized journal after a substantial revision.

Reviewer #4 (Remarks to the Author):

In this work, the authors propose a concept of twisted thermotics in a diffusive system. Since heat diffusion neither possesses the dispersion like photons nor carries the band structure as electrons, it's interesting to spot the magic angle phenomenon in a diffusion system. However, there are some points that the author must check and correct. I recommend this manuscript for publication in Nature Communications after the following comments have been addressed.

(1) The title is so general that the reader can not get much information from it. The twisted thermotics can also refer to the concepts of thermal radiation “Phys. Rev. B. 103, 155404 (2021), Phys. Rev. B. 103, 235415 (2021)” and phonon thermal conductivity of other systems “Nature. 97, 660-665 (2021)”. The author should change the title.

(2) Authors should avoid overly lengthy introduction. The overly long introduction would make it difficult for readers to quickly capture the core of the work.

(3) As presented in Fig. 2d, the imitated thermal magic angle can be constructed when the practical structure parameters are considered. Since the imitated thermal magic angle is referred to the maximum point of the temperature gradient, it is obvious to see that the imitated thermal magic angle might disappear when the structure parameter (w) becomes larger. It would be helpful if the authors make some clarifications or statements on this issue.

(4) The authors provide a general framework of achieving tunable effective thermal conductivity tensor via twisting specific angles, i.e., Eq. (1). It's obvious that the anisotropically effective thermal conductivities can be reserved, which is different from that proposed in [Nat. Mater. 18, 48 (2019)]. The authors should clarify this point.

(5) The authors have made the general expression of achieving the heat flux bending angle. This aspect is quite important to achieve the thermal rotation effect as shown in Fig. 4. The authors should provide the related statements in the main contents to highlight the significance of heat flux bending angle.

(6) Considering the 3D printing and graphene coatings used in this work, the effects of thermal contact resistance might be neglected with the current implementations. However, the authors have not made such a statement. The authors should clarify this point to improve the strictness.

(7) Can the scheme be applied for non-uniform boundary conditions? Some discussions can be added about it.

Reviewer #1 (Remarks to the Author):

The authors describe theoretical and simulation work on a structure with two thermally conductive layers twisted by an angle relative to each other. The authors demonstrate some interesting thermal functionalities that emerge out of this twisted system. However, the work is done entirely in the diffusive regime, assuming heat the heat diffusion equation and Fourier law, as implemented in the commercial tool COMSOL, are valid. The dimensions of the stripes in the two layers are not specified so it is impossible to say whether they are macroscopic and HDE is valid or if they are microscopic and HDE is not valid. The experimental work seems to indicate the former. However, the experimental work does not appear to be on striped layer but rather a circular cloaking setup with equally spaced holes in the material.

Our reply: We thank the reviewer for the accurate summarization and comments on our work. These comments and suggestions are all helpful and valuable for improving our manuscript. We have carefully considered the comments with the detailed responses listed below.

Firstly, the reviewer is right that our work is at the macroscopic scale where HDE is valid. Even though, intriguing twist processes themselves are not limited to twisted bilayer structures at the microscopic scale. We have first proposed a theoretical and macroscopic twisted mechanism in a diffusion system. As we have said in the main text, heat diffusion is intrinsically different from microscopic electron transport and photonic couplings, and further it is ambiguous how to design any thermal analogue of magic angle. The electrical conductivity tensor of twisted bilayer graphene has an inflection point^{1,2} at a certain small magic angle. Thus, inspired by the similarity between electrical and thermal conductivity tensors, we have subtly designed a similar inflection point of thermal conductivity tensor at a small angle when the stripes with the same thermal conductivity are alternately arranged on two layers as shown in Fig. R1.

Fig. R1 The curves of effective thermal conductivity κ_{xx}^{eff} along the x direction of two striped layers.

Further, the experimental work demonstrates that the thermal function switching from cloaking to concentration can be realized by judiciously twisting an analog thermal magic angle for the circular stripe layers (Fig. 3a) in the main text. The thermal function switching results from the circular stripe bilayers, not the equally spaced holes in the material which are just used to obtain an effective conductivity.

The other large issue with this manuscript for me is that the authors use the analogy to twisted bilayer graphene throughout and there really is no connection whatsoever. TBG creates flat bands microscopically through interlayer interactions and the Moire pattern. The flatness of the bands is crucial and emerges from the twist angle. Here the authors do not explicitly consider any coupling between the two layers as a function of the angle. Further, they state that "heat diffusion neither possesses the dispersion like photons nor carries the band structure as electrons, the concept of magic angles in twistrionics or opto-twistrionics is absent for conductive heat transfer". This is strictly only true in the diffusive regime. Heat transfer can also be hydrodynamic, which has been demonstrated in graphene, and then it has a dispersion that can be impacted by twist. Further, at the microscopic scale, phonons possess a dispersion and it has been demonstrated to be impacted by twist. Therefore, the limitation arises out of the assumption that heat is purely diffusive, which is only valid at macroscopic scales. Thus this work has very little to do with twisted bilayer materials.

Our reply: Thanks for carefully raising this issue and giving suggestions. Referee #1 comments that we cannot use the analogy to twisted bilayer graphene due to the lack of the flat bands induced by the moiré pattern in diffusion process, and thus there seems to be no connection. However, as the concept of "twist" is introduced to other fields such as photonics³⁻⁵ and acoustics⁶, the definition of the magic angle is also being broadened. In electronic systems, the magic angle arises from the strong interaction between particles and nontrivial band topology physics. The resulting flat band structure near charge neutrality exhibits a half-filled insulating phase at zero magnetic fields, suggesting a Mott-like insulator arising from electrons localized in the moiré superlattice. In the photonics system, the magic angle corresponds to the dispersion flat band of the momentum space related to a single energy, realizing the topological transition of dispersion from close to open. The underlying mechanism is the mutation point of the optical conductivity tensor of the bilayer system. Similarly, in our macroscopic twisted diffusion system, we also use this kind of analogous principle to enable the control and manipulation of the conductive heat behaviors from concentrator to cloak.

When it comes to the coupling at the interface of two striped layers, as shown in Fig. S7a from the Supporting Information (SI), we have reconstructed a strong heat exchange coupling at the original state (zero degrees) via a dislocation arrangement of stripes at two layers, and it can change the main diagonal item $\kappa_{1,xx}^{\text{eff}}$ largely through the symmetric twisted manipulation from zero degrees to the analog thermal magic angle. After

surpassing the thermal magic angle, the coupling in the main diagonal item $\kappa_{1,xx}^{\text{eff}}$ mainly vanishes, because it is almost same to the main diagonal item $\kappa_{1,xx}^{\text{eff}}$ in only one stripe layer without coupling. As a result, it is easy to observe a cloaking effect (Fig. 3b) at zero degrees in the coupled stripe bilayer while we can only acquire thermal concentration in decoupled stripe bilayer (Fig. S7b), indicating the transformation of a heat exchange coupling from strong to weak.

We agree with Referee #1's view that hydrodynamic process can provide a dispersion that can be impacted by a twist. However, these designs inevitably involve heat convection, like our previous works that can realize the concept of anti-parity-time symmetry⁷ and Weyl exceptional ring⁸. Similarly, twist-induced anomalies have also been observed in phonon hydrodynamics in microscopic systems. However, the analogue concept of twisted magic angle remains unexplored in pure conduction regime.

Therefore, although our work does not strictly correspond to twisted bilayer materials in the traditional sense, we have first introduced a new phenomenon of varied thermal conductivity tensor in a twisted diffusion system and in turn it can expand the potential universal applicability of twistrionics.

Reviewer #2 (Remarks to the Author):

The authors report an interesting work related to twisted material systems. The manuscript discusses the existence of a thermal analogue of the magic angle, identified with the term twisted thermotics in a twisted diffusive system. The proposed methodology is of interest for the readership of twisted-systems and materials containing moiré-superlattices.

Our reply: We thank the referee for the positive and constructive remarks. These comments and suggestions are all valuable and helpful for improving our manuscript. We have carefully considered the comments with the detailed responses listed below:

The method presents the concept of imitated thermal magic-angle. Can the authors better explain this concept in the manuscript?

Our reply: We thank the referee for carefully raising this issue and giving suggestions. We apologize for the unclarity of imitated thermal magic-angle in the original manuscript. We have included the discussion in the revised manuscript (page 5) and the details are shown listed below:

“Herein, we have first introduced the two-line moiré patterns via twisting stripe bilayers into a diffusive system to macroscopically modulate the heat exchange coupling as a function of the angle. As we know, twisted bilayer graphene can modulate the conductivity as a function of the angle through the formed moiré patterns and interlayer interactions, resulting in a magic-angle phenomenon. Interestingly, in our proposed diffusion system, we can utilize the geometry change of formed two-line moiré patterns to macroscopically modulate the thermal conductivity tensor as a function of the angle. Moreover, as shown in Fig. S2, we can further acquire a similar inflection point of thermal conductivity along the uniaxial direction at a small angle through the reconfigurable permutations of stripes in a diffusive system as an analogy to the magic angle phenomenon of conductivity tensor in the twisted bilayer graphene.”

In Fig. 1A the twisted stripe system is not clear. Does it refer to a twisted-graphene stripe or to a more general system. It would be helpful for the reader to see a direct applicability of the method.

Our reply: We thank the referee for pointing out this issue. We are sorry that the unclarity of the twisted stripe system might have brought certain confusion to the referee. In Fig. 1A, the twisted stripe system for diffusion at macroscopic scale refers to a more general system, not just a twisted-graphene stripe. Especially,

for thermal management applications, this method can be used to design a thermal magic angle, resulting in a function switching from cloaking to concentration.

When considering twisted 2D-systems with certain chemical composition, would this method be still applicable?

Our reply: Thank you for raising this good question. Generally, our twisted 2D-systems with certain chemical compositions at the macroscopic scale should still construct a similar thermal magic angle related to the structure parameters of stripes. However, currently, there is a prerequisite that the effective thermal conductivities of twisted 2D-systems with certain chemical composition at macroscopic scales should not rely heavily on the temperature. That is, temperature changes should have a neglectable influence on the effective thermal conductivities of these twisted systems. If the temperature has a strong influence on the effective thermal conductivities of twisted 2D-systems with certain chemical composition, we need to consider the effect of temperature before designing a similar thermal magic angle and it may utilize Boltzmann transport theory with first-principles calculations under the framework of density functional theory. For example, one recent literature has reported thermal conductivity in intermetallic clathrates related to temperature and the corresponding equations are shown as follows⁹:

$$\kappa = \kappa_e + \kappa_l \quad (1)$$

$$\kappa_l = \frac{1}{\Omega} \sum_{iq} g_q v_{iq} \otimes v_{iq} \tau_{iq} c_{iq} \quad (2)$$

$$\kappa_e = \kappa^0 - S^2 \sigma T \quad (3)$$

where κ_l , κ_e , and κ are the lattice thermal conductivity, the thermal conductivity related to electrons, and total thermal conductivity in intermetallic clathrates, respectively. Ω is the volume of a unit cell, g_q is the \mathbf{q} -point weight, and v_{iq} is the group velocity of mode i at point \mathbf{q} of the Brillouin zone. τ_{iq} and c_{iq} are the relaxation time and mode-specific heat capacity, respectively. S , σ , and T are the Seebeck coefficient, the electronic conductivity and temperature, respectively.

Thus, a similar thermal magic angle related to both temperature and structure parameters of stripes might be theoretically derived based on the above equations. Therefore, this method could be still feasible, and we look forward to extended works on this related topic.

Would the stripe-fabrication affect the overall crystallinity of samples?

Our reply: Thank you for the question. Yes, the stripe-fabrication will have a certain influence on the overall crystallinity of samples. As we know, chemical vapor deposition (CVD) is still an effective way for the

preparation of monolayer graphene with a large area and high quality. However, in our twisted systems, the graphene is mainly used to accelerate the formation of steady-state thermal coupling phenomena and the cost of CVD is expensive. Thus, we adopt a low-cost electronic spraying technique to fabricate the graphene coatings on the surface of the 3D printed aluminum stripes. The overall crystallinity of samples will affect the diffusion velocity and further result in a longer time to arrive at the formation of steady-state thermal coupling phenomena. We believe the effect of the overall crystallinity of samples under different stripe-fabrication conditions is interesting in twisted systems at macroscopic scales and look forward to extended works on this related topic.

In twisted graphene, van Hove singularities can be significantly altered or weakened by the twist-disorder parameter or by defects, this makes reproducibility extremely challenging. Can the authors provide practical examples of as fabricated twisted stripe-systems, with scanning tunneling microscopy or transmission electron microscopy? Would this type of system still exhibit van Hove singularities, can the authors provide an example?

Our reply: We thank the referee for raising this concern and giving the useful suggestions. It is easy to reproduce the analog thermal magic angle phenomenon in our twisted stripe-systems due to the nonexistence of van Hove singularities^{10,11} in twist-disordered graphene coatings. One related literature has reported that structural disorder can wash out Van Hove singularities¹². In our twisted system, the graphene coatings fabricated by an electronic spraying technique have considerable twist-disorder microstructures from the SEM morphology as shown in Fig. R2. Therefore, we believe that van Hove singularities of graphene coatings have been totally washed out by these twist-disorder parameters. Additionally, the graphene coatings are mainly used to accelerate the formation of the steady-state thermal coupling phenomenon in our twisted systems.

Fig. R2 The cross-sectional SEM images (a, b and c) of a cut-off sample with graphene coatings.

At the macroscopic scale, van Hove singularities in our twisted system cannot occur as we said that “heat diffusion neither possesses the dispersion like photons nor carries the band structure as electrons, the concept of magic angles in twistrionics or opto-twistrionics is absent for conductive heat transfer”. However, just one piece of twisted graphene for rotation angles θ of less than about 3.5° might still microscopically possess the van Hove singularities while the van Hove singularities do not exist in our macroscopic diffusion system for

heat diffusion. Thus, currently, there is not a direct connection between macroscopic heat diffusion phenomena and van Hove singularities. However, the van Hove singularities proposed by the referee is interesting, and they might have some influence on the thermal diffusion in twisted systems at the microscopic scale, which is out of the scope of this paper. Therefore, we look forward to extended works on this related topic.

Reviewer #3 (Remarks to the Author):

Motivated by the emerging field of twistronics, this work proposes the concept of “twisted thermotics”, which seems interesting at first glance. However, after carefully going through the manuscript, I think there are quite a few important issues to be clarified or resolved before it can be considered for publication. In particular, there are many writing problems which make the manuscript fairly hard to follow.

Our reply: We thank the referee for the constructive remarks. These comments and suggestions are all valuable and helpful for improving our manuscript. We have carefully considered the comments with the detailed responses listed below:

Below are some of my major concerns/questions.

1. About the basic idea

- What is twisted thermotics? Is it a device or a field?

Our reply: We thank the referee for pointing out this problem. The term “**thermotics**” refers to the thermal field and it has been used by several research works¹³⁻¹⁵. We understand that this term may be controversial and cause inconvenience in understanding. Thus, we change the title of the article from “Twisted Thermotics” to “Twisted Conductive Bilayer”.

- What makes a diffusive system special in the study of twist-angle-induced effects?

Our reply: Thank you for raising this issue. Heat diffusion neither possesses the dispersion like photons nor carries the band structure as electrons. Though convection could be a helping hand to some extent for the sake of establishing the band structure, the concept related to magic angles is absent for purely conductive heat transfer. So far, the general thermal coupling mechanism related to twist-angle-induced effects does not have been reported in diffusive systems. As we know, opto-twistronics expands the potential applicability of twistronics from electrons to photonics. Furthermore, inspired by the emerging field of twistronics, we have firstly proposed a theoretical and general twisted thermotics theory in twisted systems at the macroscopic scale and the discovered magic-angle phenomenon in heat diffusion is rather unexpected and novel.

- What is the thermal magic angle? What makes it magic?

Our reply: We thank the referee for asking these two good questions. We are sorry that the thermal magic angle is not illuminated clearly in the original manuscript, and it might bring some confusion to the referee. In

our twisted systems, we have achieved a similar inflection point of the thermal conductivity tensor by twisting two conductive layers with a small angle based on a certain arrangement of stripes. This twisted angle is defined as the thermal magic angle. The electrical conductivity tensor of twisted bilayer graphene has an inflection point^{1,2} at a certain small magic angle. Meanwhile, given the similarity between electrical and thermal conductivity tensors, we name the dramatic change of thermal conductivity tensor at a special angle as thermal magic angle, as a counterpart of the electrical conductivity tensor^{1,2}. The definition of magic angles might be a historical issue. The magic angle in electronic systems originates from strong interaction among particles and nontrivial band topology physics. The resulting flat band structure near charge neutrality exhibits half-filling insulating phases at zero magnetic field, which shows to be a Mott-like insulator arising from electrons localized in the moiré superlattice. The photonic magic angle corresponds to the photonic dispersion flat band in momentum space³⁻⁵. By twisting the bilayer anisotropic system, the hybridized dispersion undergoes a topological transition from closed to open. The critical point of this transition corresponds to the magic angle. Similarly, twisted bilayer acoustic superlattices also exhibit intriguing properties. Through the coupling of bilayer acoustic high-order insulators, the system exhibits unprecedented topological indices and layer-hybridized corner states⁶. Those important works hold the similar argument like ours. Herein, the thermal magic angle in our twisted systems makes transportation of diffusion to be also “magic” due to the dramatic change of thermal conductivity tensor at the thermal magic angle. This is analogous to the magic angles in twisted bilayer graphene where the electrical conductivity tensor heavily changes^{1,2}. We have included more descriptions of thermal magic angle in the revised manuscript (page 5). Besides, to investigate the evolution of thermal magic angle phenomena along with the increase of layer number, a twisted trilayer diffusive system was numerically carried out and the evolutions of the temperature distribution and gradient are shown in Fig. S9.

• Why is there a thermal magic angle in the proposed system? How are the two layers of stripes coupled? How does their coupling vary with the twist angle?

Our reply: Thank you for pointing out these issues. Practically, in the currently proposed system, a thermal magic angle occurs under a certain arrangement of not-large-enough stripes on two layers. Specifically, for physicists, we cannot change the general form of the derived thermal conductivity tensor when the stripes are large enough so as to ignore the influence of structure parameters. However, we can change the arrangement of stripes to construct a different thermal conductivity tensor at the original twist angle and it should still possess the general form of the thermal conductivity tensor when the stripes are large enough. Thus, if two kinds of stripes are alternately arranged on two layers as shown in Fig. R1, we can acquire a counterintuitive and different thermal conductivity tensor which is expressed as Eq. (S34) in the Supporting Information (SI) under the conditions that the stripes are large enough. Obviously, considering the practical applications of the

twisted diffusion system, the stripes cannot be large enough and the influence of the structure parameters should be theoretically taken into account. When the direction of heat flux is along the positive direction of x axis, it is easy to find out that twisting the practical stripes to form a zigzag structure at a small angle can maximumly reduce the heat diffusion. Meanwhile, from the simulation results (Fig. 1d) in the main text, the influence of the structure parameters induces a gradient effect from a maximum value at the original angles to a minimum value after twisting a small angle related to structure parameters. Eventually, there exists a thermal magic angle induced by the influence of structure parameters in our proposed diffusion system.

Then, the two layers of stripes are coupled by heat exchange, and we are interested in the change of potential heat exchange coupling strength in the two layers of stripes via twisted manipulations. First of all, for any stripe, we need to calculate the theoretical expression of the effective thermal conductivity tensor based on the transformation thermodynamics theory. When the two layers of stripes are aligned, we can address the heat exchange coupling in the two layers of stripes with a general framework of the effective thermal conductivity tensor based on the effective medium theory. Meanwhile, we twist the two stripe moiré patterns to modulate the effective thermal conductivity tensor so as to tailor the coupling change as a function of the angle. Compared with the phenomena in the decoupled stripe bilayer (Fig. S7b-d), the heat exchange coupling strength (Fig. S7a) will become weaker and weaker before reaching the thermal magic angle value, and it only keeps the coupling mainly originating from the off-diagonal items of effective thermal conductivity tensor after surpassing the thermal magic angle via twisted manipulations. Thus, the thermal magic angle also corresponds to the inflection point of the heat exchange coupling strength in a twisted system.

• What is the relation between the thermal magic angle here and the magic angles of twisted bilayer graphene?

Our reply: Thank you for carefully raising this issue. Strictly, the relation is that the thermal magic angle here in our twisted systems learns from the magic angles of twisted bilayer graphene under the similarity between electrical and thermal conductivity tensors. Specifically, twisted graphene bilayers can modulate the conductivity tensor as a function of the angle through the formed moiré patterns and interlayer interactions, resulting in a magic-angle phenomenon. Interestingly, in our proposed diffusion system, we can similarly utilize the geometry change of formed two-line moiré patterns to macroscopically modulate the thermal coupling as a function of the twist angle, leading to a spatial reconfiguration of the thermal conductivity tensor. Moreover, as shown in Fig. S2, we can further acquire a similar inflection point of thermal conductivity tensor at a small angle through the reconfigurable permutations of stripes in a diffusive system as an analogy to the inflection point of conductivity tensor in twisted bilayer graphene. In all, we have first introduced the stripe bilayers into a diffusive system to macroscopically modulate the thermal coupling as a function of the twist angle through the formed two-line moiré patterns at stripe bilayers, and it has not been reported in previous diffusion systems. Last but not least, the thermal magic angle is a bit more unexpected because heat diffusion

neither possesses the dispersion like photons nor carries the band structure as electrons. The novelty is more pronounced when we discover magic-angle related phenomena in purely conductive heat transfer.

2. About the theoretical/numerical study

- Most of the subscripts are not properly explained.

Our reply: We sincerely thank the reviewer for careful reading. We apologize for the inappropriate explanation of most subscripts. To solve this issue, we have carefully revised our manuscript and added more proper explanations in the Supplementary Information (SI).

- Many of the symbols and formulas are not easy to follow.

Our reply: We are sorry that many symbols and formulas bring a certain difficulty to follow for the referee. To improve this situation, we have carefully revised these symbols and formulas in both the main text and SI.

- Some of the derivations in the SI should probably go to the main text.

Our reply: Thank you for giving the useful suggestion. We are not sure which derivations in the SI should be moved to the main text. According to our humble judgments, we moved the derivations $\kappa \nabla^2 T = 0$,

$\kappa_{\text{upper}}^{\text{eff}} |^{\alpha} = \frac{J_1 \kappa_{\text{upper}}^{\text{eff}} J_1^T}{\det(J_1)}$, $\kappa^{\text{eff}} = \begin{pmatrix} \kappa_{xx}^{\text{eff}} & \kappa_{xy}^{\text{eff}} \\ \kappa_{yx}^{\text{eff}} & \kappa_{yy}^{\text{eff}} \end{pmatrix}$, $\theta_{\text{magic}} |_{\text{rec}} \approx \arcsin\left(\frac{w_I + w_{II}}{L}\right)$, $\theta_{\text{magic}} = \arcsin\left(\frac{\pi w}{2R_1}\right)$ in the SI to the

main text.

- The functional regions are not clearly marked in Fig. S2. An appropriate schematic of the device in a main figure may help.

Our reply: Thank you for providing the constructive suggestion. We have carefully provided an appropriate schematic of the device (Fig. 3a) in the revised manuscript (page 18).

3. About the experimental measurements

- Materials
 - o What is “aluminum material”?
 - o What kind of graphene is used for the coating?
 - o What is the role of the thin polypropylene film?

o How about the surface roughness?

Our reply: Thank you for pointing out these issues. We apologize for not providing the accurate type of aluminum material and graphene. We have carefully revised and solved these issues in the revised manuscript (lines 23-25, page 11). Since the thin polypropylene film has a high emissivity (0.97), its role is to isolate convection of the air and make the IR camera have an accurate measurement of thermal coupling phenomena when the thin polypropylene film is covered on the whole surface of the device. According to the related literature¹⁶, the surface roughness might have a negligible influence on the measurement of macroscopic heat exchange coupling phenomena. We have carefully revised and added these illuminations in the revised manuscript (lines 2-3, page 12).

• How are the fixed-temperature boundary conditions applied?

Our reply: Thank you for the useful question. The fixed-temperature boundary conditions are applied on the left side and right sides of this device as shown in Fig. 3a in the main text. The left side is contacted with a hot source and the right side is connected with a cold source, leading to a steady-state high temperature boundary and low temperature boundary, respectively.

• What are the uncertainties for measuring the temperature distributions?

Our reply: Thank you for pointing out this issue. In our twisted system, the uncertainties for measuring the temperature distributions are mainly from the system errors in the measurement of the IR camera and the convection of the air.

• “Herein, the measurement of effective thermal conductivity is acquired by a minimum mean-square error”
What does this mean?

Our reply: Thank you for bringing out this good issue. For the layers of stripes, to accurately obtain numerical values of effective thermal conductivity under different twist angles, we adopt a minimum mean-square error to achieve the numerical values of effective thermal conductivity in COMSOL Multiphysics. The numerical results of effective thermal conductivity in turn can further demonstrate the validity of the derived theoretical effective thermal conductivity based on transformation thermodynamics and effective medium theories.

• Is there an interfacial thermal resistance between the two layers? Does it matter?

Our reply: Thank you for the useful questions. Since 3D printing technology is used to fabricate the whole device, interfacial thermal resistance does not exist between the two layers. For the other fabrication technologies, an interfacial thermal resistance at the interface of the two layers may exist and it should have a certain influence on the thermal coupling phenomena^{17,18}. We have carefully revised and added these illuminations in the revised manuscript (lines 25-29, page 11). The interfacial thermal resistance under different stripe-fabrication conditions might introduce some interesting phenomena in twisted systems at the macroscopic scale and we look forward to extending our works on this related topic in the future.

My minor concerns are mostly with the writing. There are many typos, grammatical issues, and inaccurate expressions, some of which are listed below as examples.

Our reply: We appreciate the referee for reminding us. Following the suggestions, we have corrected them one by one.

- Please try to limit the use of phrases like “quite challenging”, “quite numerous”, “highly nontrivial”, “significantly opens up”, “extremely tuned”. The importance of a work lies in itself, not the adjectives.

Our reply: Thank you for the useful suggestions. We have limited and revised the use of phrases in our revised manuscript and SI.

- Please pay attention to the use of “the”.

Our reply: Thank you for the useful suggestions. We have revised the use of “the” in our revised manuscript and SI.

- Please pay attention to the use of “,” before “and”.

Our reply: Thanks for the useful suggestions. We have corrected the use of “,” before “and” in our revised manuscript and SI.

- “twisted modulations” and “twisted manipulations” are not grammatically sound.

Our reply: Thank you for the useful suggestions. We have corrected the use of “twisted modulations” in the introduction section (lines 4-5, page 4), and the use of “twisted manipulations” has remained unchanged (line 7, page 4).

- “it may waste the precious time”

Our reply: Thank you for the useful suggestions. We have deleted “it may waste the precious time” in the introduction section (page 3).

- “extreme thermal anisotropic”

Our reply: Thanks for the useful suggestions. We have changed “extreme thermal anisotropic” to “extremely anisotropic” in the introduction section (line 18, page 3).

- “the incomprehensibility of any thermal magic angle”

Our reply: Thanks for the useful suggestions. We have changed “the incomprehensibility of any thermal magic angle” to “the difficulty of designing any thermal magic angle” in the introduction part (line 23, page 3).

- “making it be expanded and generalized”

Our reply: Thanks for the useful suggestions. We have deleted “making it be expanded and generalized” in the introduction part (page 4).

- “via twisting twisted α -MoO₃ bilayers”

Our reply: Thanks for the useful suggestions. We have changed “via twisting twisted α -MoO₃ bilayers” to “via twisting α -MoO₃ bilayers” in the revised manuscript (line 17, page 4).

- “twisted bilayer stripe”

Our reply: Thanks for the useful suggestions. We have changed “twisted bilayer stripe” to “twisted stripe bilayers” in the revised manuscript (line 18, line 23, line 26, page 4).

- “the relative thickness of ultra-thin wall”

Our reply: Thanks for the useful suggestions. We have changed “the relative thickness of ultra-thin wall” to “the thickness of the ultrathin wall” in the revised manuscript (line 22, page 4).

- “enlarge the relatively effective thermal conductivity”

Our reply: Thanks for the useful suggestions. We have changed “enlarge the relatively effective thermal conductivity” to “enlarge the effective thermal conductivity” in the revised manuscript (line 22, page 4).

- “super high-low structure”

Our reply: Thanks for the useful suggestions. We have deleted “super high-low structure” in the revised manuscript (line 22, page 4).

- “synthesize another freedom”

Our reply: Thanks for the useful suggestions. We have changed “synthesize another freedom” to “provide another degree of freedom” in the revised manuscript (line 26, page 4).

- “upper and under layers”

Our reply: Thanks for the useful suggestions. We have changed “upper and under layers” to “upper and lower layers” in the revised manuscript (line 2, page 5; line 19, page 9; lines 15-16, page 11) and SI (line 14, page 12).

- “the down layer”

Our reply: Thanks for the useful suggestions. We have changed “the down layer” to “the lower layer” in the revised manuscript (line 26, page 7) and SI (line 15, page 12).

- “imitated thermal magic angle”

Our reply: Thanks for the useful suggestions. We have revised the use of “imitated thermal magic angle” in our revised manuscript and SI.

- “rendered a similar conception”

Our reply: Thanks for the useful suggestions. We have revised the use of “rendered a similar conception” in our revised manuscript (line 19, page 5).

- “Now, considering there is just a twisted bilayer system shown in Fig. S2. Then, we can easily obtain the general solution and matching function of this system”

Our reply: Thanks for the useful suggestions. We have changed “Now, considering there is just a twisted bilayer system shown in Fig. S2. Then, we can easily obtain the general solution and matching function of this system” to “Now, considering there is just a twisted bilayer system shown in Fig. S6, we can easily obtain the general solution and matching function of this system (see Supplementary Note 4).” in our revised manuscript (lines 19-20, page 7).

- “can be achieved between the interface of the two layers”

Our reply: Thanks for the useful suggestions. We have changed “can be achieved between the interface of the two layers” to “can be achieved through the heat exchange coupling at the interface of the two stripe layers” in our revised manuscript (lines 24-25, page 7).

- The expression of k_x

Our reply: Thanks for the useful suggestions. We have corrected “The expression of k_x ” in our revised manuscript and SI.

- The use of brackets (())

Our reply: Thanks for the useful suggestions. We have corrected “The use of brackets (())” in our revised manuscript and SI.

- “for the region III, it is composed of”

Our reply: Thanks for the useful suggestions. We have deleted “for the region III, it is composed of” in our revised manuscript (page 8) and changed “for the region III, it is composed of” to “assuming that region III is composed of” in the SI (line 10, page 13).

- “Especially, for the twisted model”

Our reply: Thanks for the useful suggestions. We have deleted “Especially, for the twisted model” in our revised manuscript (page 8) and changed “Especially, for the twisted model” to “For the twisted metadvice” in the SI (line 14, page 13).

- “arbitrarily thermal conductivities”

Our reply: Thanks for the useful suggestions. We have changed “arbitrarily thermal conductivities” to “arbitrary thermal conductivity” in our revised manuscript (line 15, page 8).

- “realization of a twisted cloak-to-concentrator”

Our reply: Thanks for the useful suggestions. We have changed “realization of a twisted cloak-to-concentrator” to “realization of a cloak-to-concentrator” in our revised manuscript (line 4, page 9).

- “which all the structures with only the aluminum material”

Our reply: Thanks for the useful suggestions. We have deleted “which all the structures with only the aluminum material” in our revised manuscript (line 7, page 11).

- “secondary outer”

Our reply: Thanks for the useful suggestions. We have changed “secondary outer” to “second outer” in our revised manuscript (line 8, line 11, page 11).

- “as shown in below” in the SI

Our reply: Thanks for the useful suggestions. We have changed “as shown in below” to “as shown below” in our revised SI (line 18, page 10).

Overall, I am afraid this work does not meet the high standards of Nature Communications, and may be suitable for a more specialized journal after a substantial revision.

Our reply: We are grateful for the referee's critical comments. We have taken all the suggestions into account and substantially advanced the work. We wish to highlight the novelty once again that we propose a first twisted mechanism in macroscopic twisted diffusion systems and validate our concept in experiments. We added two new magic-angle phenomena in twisted systems by increasing the layer numbers and introducing an elliptic boundary condition as Referee #4 suggested. Both have not been reported in previous diffusive systems and further strengthen the significance of our work. The extension capability of our design under an elliptic boundary condition has been verified with the numerical results. The potential twisted mechanism of other moiré patterns, such as tilted moiré patterns can also be developed as promising extensions of twisted thermodynamics. These new phenomena and extensions, we hope, have addressed the referees' concerns. In short, we believe that we are introducing interesting new physics that is surprisingly unexpected and has been never reported before.

Reviewer #4 (Remarks to the Author):

In this work, the authors propose a concept of twisted thermotics in a diffusive system. Since heat diffusion neither possesses the dispersion like photons nor carries the band structure as electrons, it's interesting to spot the magic angle phenomenon in a diffusion system. However, there are some points that the author must check and correct. I recommend this manuscript for publication in Nature Communications after the following comments have been addressed.

Our Reply: We thank the referee for positively commenting that the concept of twisted thermotics in our twisted system is novel and magic angle phenomenon is interesting. These comments and suggestions are all helpful and valuable for improving our manuscript. We have carefully considered the comments with the detailed responses listed below:

(1) The title is so general that the reader cannot get much information from it. The twisted thermotics can also refer to the concepts of thermal radiation “Phys. Rev. B. 103, 155404 (2021), Phys. Rev. B. 103, 235415 (2021)” and phonon thermal conductivity of other systems “Nature. 97, 660-665 (2021)”. The author should change the title.

Our Reply: Thanks for carefully raising this issue and giving suggestions. As suggested by the reviewer, we have corrected the “Twisted Thermotics” to Twisted Conductive Bilayer”.

(2) Authors should avoid overly lengthy introduction. The overly long introduction would make it difficult for readers to quickly capture the core of the work.

Our Reply: Thanks for carefully raising this issue and giving suggestions. We have tried our best to shorten the overly long introduction and make some changes marked in red in the revised manuscript (pages 2-4).

(3) As presented in Fig. 2d, the imitated thermal magic angle can be constructed when the practical structure parameters are considered. Since the imitated thermal magic angle is referred to the maximum point of the temperature gradient, it is obvious to see that the imitated thermal magic angle might disappear when the structure parameter (w) becomes larger. It would be helpful if the authors make some clarifications or statements on this issue.

Our Reply: Thanks for carefully pointing out this issue and giving suggestions. To construct a magic angle phenomenon in a twisted diffusion system, the zigzag connection in the two layers should be maintained and it is easy to observe that the least number of stripes in each layer is three as shown in Fig. R3.

Fig. R3 Schematics of the least number of stripes for the current system.

In all, we have carefully revised and added some clarifications on this issue in the revised SI (see Supplementary Note 3, page 10).

(4) The authors provide a general framework of achieving tunable effective thermal conductivity tensor via twisting specific angles, i.e., Eq. (1). It's obvious that the anisotropically effective thermal conductivities can be reserved, which is different from that proposed in [Nat. Mater. 18, 48 (2019)]. The authors should clarify this point.

Our Reply: Thanks for carefully bringing out this issue and giving suggestions. In our twisted system, there exist anisotropiceffective thermal conductivities based on a new twisted thermodynamic theory. We have carefully revised and added some statements on this issue in the revised manuscript (lines 14-16, page 6).

(5) The authors have made the general expression of achieving the heat flux bending angle. This aspect is quite important to achieve the thermal rotation effect as shown in Fig. 4. The authors should provide the related statements in the main contents to highlight the significance of heat flux bending angle.

Our Reply: Thanks for carefully raising this issue and giving suggestions. We apologize for not providing the corresponding statements on the role of the heat flux bending angle to achieve the thermal rotation effect. We have carefully revised and added some statements to highlight the significance of heat flux bending angle in the revised manuscript (line 10, page 10).

(6) Considering the 3D printing and graphene coatings used in this work, the effects of thermal contact resistance might be neglected with the current implementations. However, the authors have not made such a statement. The authors should clarify this point to improve the strictness.

Our Reply: We sincerely thank the referee for the valuable and constructive comments. We apologize for not providing the corresponding statements on how to avoid the effects of thermal contact resistance. We have carefully revised and added some statements and literature to discuss why the effects of thermal contact resistance can be neglected with the current implementations in the revised manuscript (lines 25-29, page 11).

(7) Can the scheme be applied for non-uniform boundary conditions? Some discussions can be added about it.

Our Reply: Thanks for carefully pointing out this issue and giving suggestions. It is possible to generalize our approach to non-uniform boundary conditions as shown in Fig. R4. In fact, in the Methods part of our manuscript, we have included a general description based on an elliptic boundary condition. Moreover, in the SI, we have included a detailed note for an elliptic boundary condition (see Supplementary Note 7, Fig. S10). We believe the cases with more complicated geometries in non-uniform boundary conditions might exist more interesting phenomena and look forward to future works on these topics.

Fig. R4 Numerical demonstration of the thermal magic angle phenomenon under an elliptic boundary condition. a, b Temperature distributions of twisted thermal cloak-to-concentration via twisting a magic angle. **c, d** Temperature distributions along the line ($y = 0$) in twisted systems under an elliptic boundary condition.

References

1. Rickhaus, P. *et al.* Correlated electron-hole state in twisted double-bilayer graphene. *Science* **373**, 1257–1260 (2021).
2. Polski, R. *et al.* Hierarchy of symmetry breaking correlated phases in twisted bilayer graphene. Preprint at <http://arxiv.org/abs/2205.05225> (2022).
3. Liu, H. *et al.* Controlling transition photonic band with synthetic moiré sphere. Preprint at <https://www.researchsquare.com/article/rs-2429995/v1> (2023) doi:10.21203/rs.3.rs-2429995/v1.
4. Liu, Y. *et al.* Moiré-driven electromagnetic responses and magic angles in a sandwiched hyperbolic metasurface. *Photon. Res.* **10**, 2056 (2022).
5. Wang, Y. *et al.* Observation of magic angle and wall state in twisted bilayer photonic graphene. Preprint at <https://doi.org/10.48550/arXiv.1911.09174> (2019).
6. Wu, S.-Q. *et al.* Higher-order topological states in acoustic twisted Moiré superlattices. *Phys. Rev. Applied* **17**, 034061 (2022).
7. Li, Y. *et al.* Anti-parity-time symmetry in diffusive systems. *Science* **364**, 170–173 (2019).
8. Xu, G. *et al.* Observation of Weyl exceptional rings in thermal diffusion. *Proc. Natl. Acad. Sci. U.S.A.* **119**, e2110018119 (2022).
9. Lindroth, D. O. *et al.* Thermal conductivity in intermetallic clathrates: A first-principles perspective. *Phys. Rev. B* **100**, 045206 (2019).
10. Yan, W. *et al.* Angle-dependent van Hove singularities and their breakdown in twisted graphene bilayers. *Phys. Rev. B* **90**, 115402 (2014).
11. Wang, Y., Hong, L., Wang, Y., Schirmacher, W. & Zhang, J. Disentangling boson peaks and van Hove singularities in a model glass. *Phys. Rev. B* **98**, 174207 (2018).
12. Chumakov, A. I. *et al.* Role of disorder in the thermodynamics and atomic dynamics of glasses. *Phys. Rev. Lett.* **112**, 025502 (2014).
13. Hu, R. *et al.* Illusion thermotics. *Adv. Mater.* **30**, 1707237 (2018).
14. Li, Y. *et al.* Temperature-dependent transformation thermotics: from switchable thermal cloaks to macroscopic thermal diodes. *Phys. Rev. Lett.* **115**, 195503 (2015).
15. Xu, L.-J. & Huang, J.-P. Transformation thermotics and extended theories: inside and outside metamaterials. (Springer Nature, 2023). doi:10.1007/978-981-19-5908-0.
16. Prasher, R. Thermal Interface Materials: Historical Perspective, Status, and Future Directions. *Proceedings of the IEEE* **94**, 1571–1586 (2006).
17. Chen, J., Xu, X., Zhou, J. & Li, B. Interfacial thermal resistance: Past, present, and future. *Rev. Mod. Phys.* **94**, 025002 (2022).
18. Zheng, X. & Li, B. Effect of interfacial thermal resistance in a thermal cloak. *Phys. Rev. Applied* **13**, 024071 (2020).

Reviewers' comments:

Reviewer #2 (Remarks to the Author):

The authors have addressed the questions from my previous review, however the link between the presented thermal-analogue of magic angle and twisted graphene bilayers appears still weak. Can the authors improve the explanation on existence/absence of interlayer coupling in the system that they analyze? I believe that broadening the concept of magic-angle may help the readership and audience of the journal. The introduction has to be improved, to clarify the concept of magic-angle in a broader scope. The comparison with the concept of van Hove singularity is also unclear (<https://journals.aps.org/prl/abstract/10.1103/PhysRevLett.109.196802>, <https://www.nature.com/articles/nature26154>). I believe that the manuscript needs to explain in a clearer manner the concept of magic-angle for the presented system and its implications.

Reviewer #3 (Remarks to the Author):

I appreciate the authors for the detailed responses and revisions. Unfortunately, after carefully reading all the materials, I am afraid I am still not convinced that this work is suitable for Nature Communications.

My main concern remains that I have failed to see a real analogy with the more interesting cases of electronic and photonic transport in twisted materials systems. In terms of electronic transport, the phenomena of unconventional superconductivity and correlated insulator behaviour in twisted bilayer graphene were far from obvious or easily predictable with existing theoretical models. In comparison, in the case of thermal transport presented in this work, the inflection point of the thermal conductivity as a function of the twist angle is defined as the thermal magic angle, which seems somewhat trivial instead of magic. Moreover, transport in this system can be readily modelled with commercially available software like COMSOL based on well understood physics.

With the above said, I feel some of the expressions like “these correlated diffusion systems” and “where strong coupling emerges within a twisted diffusion system” could be misleading or confusing.

Regarding the experiment, I find the sentence “Obviously, it is easy to observe that the numerical and experimental results are almost the same” inadequate as a discussion of the experimental results. More importantly, it seems that the upper and lower layers in the model are actually 3D printed into one connected piece. If this is indeed the case, then thermal transport in this system cannot be tuned by the

twist angle. It would be interesting to see the more realistic case of two independent layers separated by an interface thermal resistance, which may even dominate the transport behaviour.

In addition, despite much improvement in the writing, there is still room to do better. For example:

1. “no matter whether in electronics or photonics”
2. On one hand, the left and middle panels of Fig. 1a seem to be marginally relevant. On the other, the coordinate system used in this work may be better introduced in the main figures.
3. The four regions are still not marked correctly in Fig. S6b.
4. Scales bars should be added in Fig. S8.
5. Many of the supplementary equations seem to be somewhat redundant. It seems that some are for general geometric parameters while others are for the case of $w_I = w_{II}$.

Reviewer #4 (Remarks to the Author):

The authors have basically replied to all my concerns. I am happy about the changes made by the authors. An interesting magic angle phenomenon in a diffusion system has been discovered in purely conductive heat transfer based on a new concept of twisted thermotronics and the underlying mechanism of such a magic angle phenomenon is well explained based on systematic simulations and further experimental demonstration. The results reported in this manuscript are new and of some interest to the metamaterials containing moiré patterns and twistrionics community. In my view, the current manuscript is suitable for acceptance in terms of scientific content, and I recommend this manuscript for publication after some minor revision :

1. The title is still so generalised that the reader is left wondering what specific scientific problem the author has studied. The author should have reflected in the title that the author introduced this concept of the moiré into the thermal diffusion system.
2. It is not necessary to show the twisted graphene bilayers and twisted αMoO_3 bilayers.

Reviewer #2 (Remarks to the Author):

The authors have addressed the questions from my previous review, however the link between the presented thermal-analogue of magic angle and twisted graphene bilayers appears still weak.

Our Reply: We thank the reviewer for the accurate summarization and comments on our work. We apologize for not adequately providing the link between the presented thermal magic angle and twisted graphene bilayers. The twisted graphene bilayers can accelerate the formation of steady-state thermal coupling phenomenon in our diffusion systems through the interlayer coupling. We have carefully considered the comments with the detailed responses listed below.

Can the authors improve the explanation on existence/absence of interlayer coupling in the system that they analyze?

Our Reply: We thank the referee for carefully pointing out this issue. The basic interlayer coupling of the effective thermal conductivity tensor in the system can be described by the equation below:

$$\begin{aligned}
 \mathbf{K}^{\text{eff}} &= \begin{pmatrix} K_{xx}^{\text{eff}} & K_{xy}^{\text{eff}} \\ K_{yx}^{\text{eff}} & K_{yy}^{\text{eff}} \end{pmatrix} \\
 &= \begin{cases} \begin{pmatrix} \frac{(w_1 + w_2)(\kappa_1 + \zeta_1 \kappa_1) + \zeta_1(\kappa_1 - \kappa_1)w_2}{(w_1 + w_2)(\kappa_1 + \zeta_1 \kappa_1) - (\kappa_1 - \kappa_1)w_2} & 0 \\ 0 & \frac{w_1 \kappa_1 + w_2 \kappa_2}{w_1 + w_2} \end{pmatrix} & (\theta = \theta_2 = \frac{m\pi}{2}) \\ \begin{pmatrix} \frac{\kappa_1 [\cos^2(\theta - \alpha) + \cos^2(\theta_2 - \alpha)] + \kappa_2 [\sin^2(\theta - \alpha) + \sin^2(\theta_2 - \alpha)]}{2} & \frac{(-\kappa_1 + \kappa_2) [\cos(\theta - \alpha) \sin(\theta - \theta) + \cos(\theta_2 - \alpha) \sin(\theta_2 - \alpha)]}{2} \\ \frac{(-\kappa_1 + \kappa_2) [\cos(\theta - \alpha) \sin(\theta - \alpha) + \cos(\theta_2 - \alpha) \sin(\theta_2 - \alpha)]}{2} & \frac{\kappa_1 [\sin^2(\theta - \alpha) + \sin^2(\theta_2 - \alpha)] + \kappa_2 [\cos^2(\theta - \alpha) + \cos^2(\theta_2 - \alpha)]}{2} \end{pmatrix} & (\theta - \theta_2 \neq m\pi) \end{cases} \quad (1)
 \end{aligned}$$

Since the introduction of the graphene layer with a thickness of d_{gra} in the system, the interlayer coupling can be affected by the thermal resistance between the graphene and aluminum alloy. According to the related literature¹, the new interlayer coupling of the effective thermal conductivity tensor in the system is derived as:

$$\begin{aligned}
 \mathbf{K}_{\text{cov}}^{\text{eff}} &= \begin{pmatrix} K_{xx}^{\text{eff}} & K_{xy}^{\text{eff}} \\ K_{yx}^{\text{eff}} & K_{yy}^{\text{eff}} \end{pmatrix} \\
 &= \begin{cases} \begin{pmatrix} \frac{(w_1 + w_2)(\kappa_1 + \zeta_1 \kappa_1) + \zeta_1(\kappa_1 - \kappa_1)w_2}{(w_1 + w_2)(\kappa_1 + \zeta_1 \kappa_1) - (\kappa_1 - \kappa_1)w_2} \kappa_{\text{gra}} & 0 \\ d\kappa_{\text{gra}} + d_{\text{gra}} \frac{(w_1 + w_2)(\kappa_1 + \zeta_1 \kappa_1) + \zeta_1(\kappa_1 - \kappa_1)w_2}{(w_1 + w_2)(\kappa_1 + \zeta_1 \kappa_1) - (\kappa_1 - \kappa_1)w_2} & \frac{w_1 \kappa_1 + w_2 \kappa_2}{w_1 + w_2} \kappa_{\text{gra}} \\ 0 & \frac{d\kappa_{\text{gra}} + d_{\text{gra}} \frac{w_1 \kappa_1 + w_2 \kappa_2}{w_1 + w_2}}{d\kappa_{\text{gra}} + d_{\text{gra}} \frac{w_1 \kappa_1 + w_2 \kappa_2}{w_1 + w_2}} \end{pmatrix} & (\theta = \theta_2 = \frac{m\pi}{2}) \\ \begin{pmatrix} \frac{\kappa_1 [\cos^2(\theta - \alpha) + \cos^2(\theta_2 - \alpha)] + \kappa_2 [\sin^2(\theta - \alpha) + \sin^2(\theta_2 - \alpha)] \kappa_{\text{gra}}}{2d\kappa_{\text{gra}} + d_{\text{gra}} \left\{ \kappa_1 [\cos^2(\theta - \alpha) + \cos^2(\theta_2 - \alpha)] + \kappa_2 [\sin^2(\theta - \alpha) + \sin^2(\theta_2 - \alpha)] \right\}} & \frac{\{(-\kappa_1 + \kappa_2) [\cos(\theta - \alpha) \sin(\theta - \theta) + \cos(\theta_2 - \alpha) \sin(\theta_2 - \alpha)]\} \kappa_{\text{gra}}}{2d\kappa_{\text{gra}} + d_{\text{gra}} \{(-\kappa_1 + \kappa_2) [\cos(\theta - \alpha) \sin(\theta - \theta) + \cos(\theta_2 - \alpha) \sin(\theta_2 - \alpha)]\}} \\ \frac{\{(-\kappa_1 + \kappa_2) [\cos(\theta - \alpha) \sin(\theta - \alpha) + \cos(\theta_2 - \alpha) \sin(\theta_2 - \alpha)]\} \kappa_{\text{gra}}}{2d\kappa_{\text{gra}} + d_{\text{gra}} \{(-\kappa_1 + \kappa_2) [\cos(\theta - \alpha) \sin(\theta - \alpha) + \cos(\theta_2 - \alpha) \sin(\theta_2 - \alpha)]\}} & \frac{\left\{ \kappa_1 [\sin^2(\theta - \alpha) + \sin^2(\theta_2 - \alpha)] + \kappa_2 [\cos^2(\theta - \alpha) + \cos^2(\theta_2 - \alpha)] \right\} \kappa_{\text{gra}}}{2d\kappa_{\text{gra}} + d_{\text{gra}} \left\{ \kappa_1 [\sin^2(\theta - \alpha) + \sin^2(\theta_2 - \alpha)] + \kappa_2 [\cos^2(\theta - \alpha) + \cos^2(\theta_2 - \alpha)] \right\}} \end{pmatrix} & (\theta - \theta_2 \neq m\pi) \end{cases} \quad (2)
 \end{aligned}$$

I believe that broadening the concept of magic-angle may help the readership and audience of the journal. The introduction has to be improved, to clarify the concept of magic-angle in a broader scope.

Our Reply: We thank the referee for carefully raising this issue and giving suggestions. We apologize for not broadening the concept of magic-angle in the original manuscript. In the introduction part, we have presented an expanded view of the magic-angle concept, transcending its origins in condensed matter physics and delving into new realms of materials science, quantum information science, biophysics, and beyond. By illuminating the remarkable properties and multifaceted applications of the magic-angle, we hope to inspire researchers to embark on new journeys of exploration and encourage the readership and audience to embrace the boundless opportunities provided by this captivating phenomenon. We also believe that this broader perspective will not only enrich the scientific community's understanding of magic-angle phenomena but also foster interdisciplinary collaborations, propelling innovative research directions and technological breakthroughs. We have included the discussion in the revised manuscript (lines 1-10, page 3) and the details are shown listed below:

“However, the concept of the magic-angle possesses far-reaching implications beyond its original domain, holding great potential for diverse scientific applications. Beyond the traditional confines of magic-angle in electronics and photonics, the magic-angle concept has the potential to permeate various scientific disciplines, pushing the boundaries of knowledge and innovation. For instance, recent studies have explored the intriguing connection between magic-angle phenomena and quantum information science, highlighting the possibility of using twisted systems as platforms for quantum computing and simulation³⁹. Moreover, the magic-angle concept has also shown promise in fields such as biophysics, where the manipulation of twist angles in DNA molecules has revealed unexpected structural and functional properties^{40, 41}.”

The comparison with the concept of van Hove singularity is also unclear (<https://journals.aps.org/prl/abstract/10.1103/PhysRevLett.109.196802>, <https://www.nature.com/articles/nature26154>). I believe that the manuscript needs to explain in a clearer manner the concept of magic-angle for the presented system and its implications.

Our Reply: We thank the referee for carefully pointing out this issue and giving suggestions. As we all know, only few layers graphene has been reported that the van Hove singularity can be observed. Currently, the graphene coatings with the thickness of about 50 μm have tens of thousands of layers and thus the concept of van Hove singularity^{2,3} cannot be constructed in our current proposed twisted diffusion system at macroscopic scales. Therefore, there might be no need to carry on the scanning tunneling microscopy for the sake of the

comparison with the concept of van Hove singularity. Moreover, compared with our twisted bilayer diffusive system without graphene coatings, the graphene coatings are just to accelerate the formation of steady-state thermal coupling phenomenon in our twisted diffusive systems as shown in Fig. R1. So, the graphene coatings as twisted bilayer nanomaterials do not influence the general theoretical framework in our twisted diffusion systems. To explain the concept of magic-angle for the presented system and its implications more clearly, we have included the discussion in the revised manuscript (lines 18-20, page 5).

Fig. R1 Transient evolutions a (without graphene coatings) and b (with graphene coatings) of thermal cloak in our twisted diffusive systems at original twisted angles.

Reviewer #3 (Remarks to the Author):

I appreciate the authors for the detailed responses and revisions. Unfortunately, after carefully reading all the materials, I am afraid I am still not convinced that this work is suitable for Nature Communications.

Our Reply: We thank the reviewer for the accurate summarization and comments on our work. We have taken all the suggestions into account and substantially advanced the work. We wish to highlight the novelty once again that we propose a first twisted mechanism in macroscopic twisted diffusion systems and validate our concept in experiments. We added the detailed analysis on existence of interlayer thermal coupling in the system as Referee #1 suggested. It has not been reported in previous diffusive systems and further strengthen the significance of our work. These new phenomena and extensions, we hope, have addressed the referees' concerns. In short, these comments and suggestions are all helpful and valuable for improving our manuscript. We have carefully considered the comments with the detailed responses listed below.

My main concern remains that I have failed to see a real analogy with the more interesting cases of electronic and photonic transport in twisted materials systems. In terms of electronic transport, the phenomena of unconventional superconductivity and correlated insulator behaviour in twisted bilayer graphene were far from obvious or easily predictable with existing theoretical models. In comparison, in the case of thermal transport presented in this work, the inflection point of the thermal conductivity as a function of the twist angle is defined as the thermal magic angle, which seems somewhat trivial instead of magic. Moreover, transport in this system can be readily modelled with commercially available software like COMSOL based on well understood physics.

Our Reply: We thank the referee for carefully raising this issue and giving suggestions. We apologize for not providing the more interesting cases in the original manuscript and we have added the formation of steady-state thermal coupling phenomenon in our twisted diffusion systems (Fig. S11) in the Supplementary Information (SI). In a trivial case, we can only acquire a normal curve of the thermal conductivity κ_{xx}^{eff} as shown in Fig. R2. However, when the stripes with the same thermal conductivity are alternately arranged on two layers in another case as shown in Fig. R1, we have subtly designed an inflection point of thermal conductivity (κ_{xx}^{eff}) at a small angle. Although the transport in this system can be modelled by COMSOL, the inflection point of thermal conductivity can not be predicted by the current well understood physics and thus we propose a first twisted mechanisms in macroscopic twisted diffusion systems and validate our concept in experiments. Besides, since the introduction of the graphene layer with a thickness of d_{gra} in the system, the interlayer coupling can be affected by the thermal resistance between the graphene and aluminum alloy and

thus a new interlayer coupling of the effective thermal conductivity tensor in the system (Equation 2) has been derived.

Fig. R2 The curves of effective thermal conductivity related to two distinct permutation stripes at two layers.

With the above said, I feel some of the expressions like “these correlated diffusion systems” and “where strong coupling emerges within a twisted diffusion system” could be misleading or confusing.

Our Reply: Thanks for the useful suggestions. We have revised the use of some expressions like “these correlated diffusion systems” and “where strong coupling emerges within a twisted diffusion system” in our revised manuscript (line 19, page 3; line 2, page 4).

Regarding the experiment, I find the sentence “Obviously, it is easy to observe that the numerical and experimental results are almost the same” inadequate as a discussion of the experimental results. More importantly, it seems that the upper and lower layers in the model are actually 3D printed into one connected piece. If this is indeed the case, then thermal transport in this system cannot be tuned by the twist angle. It would be interesting to see the more realistic case of two independent layers separated by an interface thermal resistance, which may even dominate the transport behaviour.

Our Reply: Thank you for bringing out these good questions and giving suggestions. We have added extra discussions in the experimental results (lines 27-29, page 9; lines 1-7, page 10). Furthermore, we would like to emphasize the novelty of our work once again. Our contribution lies in the introduction of a twisted mechanism within macroscopic twisted diffusion systems, a concept that has not been explored previously. Moreover, we have successfully validated this concept through experimental verification, marking the first instance of its implementation. Currently, the upper and lower layers in the practical model are indeed 3D printed into one connected piece. The integration of 3D printing technology with our concept provides a powerful means of materializing our ideas and exploring the magic angle in thermal science. By leveraging the capabilities of 3D printing, we can fabricate intricate structures with precise twist angles, allowing us to investigate and analyze the unique thermal properties associated with the magic angle. The process of 3D printing enables us to translate our theoretical understanding into physical objects, providing tangible evidence of our concept's validity.

However, it is essential to recognize the inherent limitations of 3D printing and the static nature of the printed structures. Once a design is printed, it becomes fixed and cannot be altered. In the field of photonics, similar cases arise, where fabricated structures cannot be twisted once they are completed^{4,5}. However, it is possible to fabricate new structures if the twisted angles or requirements change. This implies that, for the time being, our proof-of-concept is based on a single connected piece only, and further advancements and refinements may be extended in the future work.

While our theory provides valuable insights into which twist angles should be designed and printed to achieve the desired thermal effects, we must acknowledge that real-world implementation may not be flawless. Imperfections can arise due to various factors, such as material properties, manufacturing tolerances, or other practical considerations. These limitations remind us that our current implementation may not be perfect, and there is room for further development and optimization.

Although there exist the imperfections and limitations of our current approach, it motivates us to explore new avenues for enhancing the fabrication process and potentially incorporating dynamic elements that allow for adjustments or adaptations after 3D printing. These future advancements could expand the scope and applicability of our concept and pave the way for even more remarkable discoveries in thermal science.

In conclusion, while 3D printing serves as a valuable tool for realizing our concept and our theory guides us in designing and printing structures with specific magic angles, we acknowledge that perfection is not always achievable in practice. We recognize the static nature of printed structures and the need for further advancements to address current limitations. By embracing these challenges and actively seeking improvements, we can continue to push the boundaries of what is possible and refine our understanding of the magic angle in thermal science.

In addition, despite much improvement in the writing, there is still room to do better. For example:

Our Reply: We sincerely thank the reviewer for careful reading. We have corrected grammatical errors in our revised manuscript and the details are shown listed below:

1. “no matter whether in electronics or photonics”

Our Reply: Thanks for the useful suggestions. We have changed “no matter whether in electronics or photonics” to “no matter in either electronics or photonics” in the introduction part (lines 11-12, page 3).

2. On one hand, the left and middle panels of Fig. 1a seem to be marginally relevant. On the other, the coordinate system used in this work may be better introduced in the main figures.

Our Reply: Thanks for the useful suggestions. As suggested by the reviewer, we have corrected and deleted the twisted graphene bilayers and twisted αMoO_3 bilayers in Fig. 1a and we also have clarified the coordinate system in this current system (page 17).

3. The four regions are still not marked correctly in Fig. S6b.

Our Reply: Thanks for carefully raising this issue and giving the useful suggestions. For the Fig. S6b in the SI, we have corrected the four regions marked in this figure (page 11, SI).

4. Scales bars should be added in Fig. S8.

Our Reply: Thanks for carefully pointing out this issue and giving the useful suggestions. We have added the scales bars in Fig. S8 (page 15, SI).

5. Many of the supplementary equations seem to be somewhat redundant. It seems that some are for general geometric parameters while others are for the case of $w_I = w_{II}$.

Our Reply: Thanks for the useful suggestions. we have simplified and deleted these redundant equations in the Supplementary Information (SI).

Reviewer #4 (Remarks to the Author):

The authors have basically replied to all my concerns. I am happy about the changes made by the authors. An interesting magic angle phenomenon in a diffusion system has been discovered in purely conductive heat transfer based on a new concept of twisted thermotics and the underlying mechanism of such a magic angle phenomenon is well explained based on systematic simulations and further experimental demonstration. The results reported in this manuscript are new and of some interest to the metamaterials containing moiré patterns and twistrionics community. In my view, the current manuscript is suitable for acceptance in terms of scientific content, and I recommend this manuscript for publication after some minor revision:

Our Reply: We thank the referee for positively commenting that the concept of twisted thermotics in our twisted system is novel and magic angle phenomenon is interesting. These comments and suggestions are all helpful and valuable for improving our manuscript. We have carefully considered the comments with the detailed responses listed below.

1. The title is still so generalised that the reader is left wondering what specific scientific problem the author has studied. The author should have reflected in the title that the author introduced this concept of the moiré into the thermal diffusion system.

Our Reply: Thanks for carefully raising this issue and giving suggestions. As suggested by the reviewer, we have corrected the “Twisted Conductive Bilayer” to Twisted Moiré Conductive Bilayer”.

2. It is not necessary to show the twisted graphene bilayers and twisted αMoO_3 bilayers.

Our Reply: Thanks for carefully pointing out this issue and giving suggestions. As suggested by the reviewer, we have corrected and deleted the twisted graphene bilayers and twisted αMoO_3 bilayers in Fig. 1a (page 17).

References

1. Lambropoulos, J. C. *et al.* Thermal conductivity of dielectric thin films. *Journal of Applied Physics* **66**, 4230–4242 (1989).
2. Brihuega, I. *et al.* Unraveling the Intrinsic and Robust Nature of van Hove Singularities in Twisted Bilayer Graphene by Scanning Tunneling Microscopy and Theoretical Analysis. *Phys. Rev. Lett.* **109**, 196802 (2012).

3. Cao, Y. *et al.* Correlated insulator behaviour at half-filling in magic-angle graphene superlattices. *Nature* **556**, 80–84 (2018).
4. Du, L. *et al.* Moiré photonics and optoelectronics. *Science* **379**, eadg0014 (2023).
5. Mao, X.-R., Shao, Z.-K., Luan, H.-Y., Wang, S.-L. & Ma, R.-M. Magic-angle lasers in nanostructured moiré superlattice. *Nat. Nanotechnol.* **16**, 1099–1105 (2021).

REVIEWER COMMENTS

Reviewer #2 (Remarks to the Author):

The authors have revised the previous version of the manuscript, with efforts made to theoretically explain the concept of interlayer coupling. While the manuscript appears to have improved in contents, especially from the theoretical perspective, I am still unable to clearly see the experimental confirmation of the theoretical descriptions, especially on the concept of interlayer coupling. It is unclear the role of the graphene layer and the possible interactions with the twisted diffusive system. The system appears of particular complexity and the concept of magic-angle in Fig.S2 is still unclear. I believe that improvements are still needed to clearly present the link between the proposed theory and the experimental result presented in Fig.S2.

Reviewer #2 (Remarks to the Author):

The authors have revised the previous version of the manuscript, with efforts made to theoretically explain the concept of interlayer coupling.

Our reply: We thank the referee for the positive and constructive remarks. These comments and suggestions are all valuable and helpful for improving our manuscript. We have carefully considered the comments with the detailed responses listed below:

While the manuscript appears to have improved in contents, especially from the theoretical perspective, I am still unable to clearly see the experimental confirmation of the theoretical descriptions, especially on the concept of interlayer coupling.

Our reply: We thank the referee for the careful review. We would like to explain more about the experimental confirmation of the theoretical descriptions. The theory proposed in this work predicts the existence of a “thermal magic angle”, characterized by an inflection point in the effective conductivity (Fig.1d). This phenomenon arises due to the twisting pattern of the alternately arranged stripes. To experimentally validate the theory, we select two representative cases in which the twisted angle is 0 and 5.6° (the magic angle) to fabricate the meta-device and conduct the experiments. The experimental results (Fig. 3) exhibit a switchable effect from cloak to concentration, demonstrating a decrease in the effective conductivity. Even though we choose only two cases for experimental simplicity, a multitude of numerical simulations (Figs. 2-4) are implemented to illustrate the inflection behavior in the effective conductivity. The underlying mechanism behind the inflection behavior is rooted in the varying interlayer coupling caused by the twisted angle. In other words, the ability of heat flux transport between the upper and lower layers varies with the geometric arrangement as the twisted angles change.

It is unclear the role of the graphene layer and the possible interactions with the twisted diffusive system.

Our reply: We thank the referee for raising this question. One important reason for incorporating the graphene layer in our meta-device is to rapidly observe the establishment of steady-state thermal transport phenomena. The inclusion of graphene significantly reduces the time required to reach a steady state, thanks to its high thermal conductivity. It serves solely to expedite the observation of the phenomena we have discovered, not affecting the physical model, observed phenomena, and resulting conclusions. In the revised Supplementary Information (Supplementary Note 8), we provide additional analysis on its impact on this diffusive system, which indicates that the effective thermal conductivity can be enhanced, and the temperature evolution can be expedited by introducing the graphene coatings.

The system appears of particular complexity and the concept of magic-angle in Fig.S2 is still unclear. I believe that improvements are still needed to clearly present the link between the proposed theory and the experimental result presented in Fig.S2.

Our reply: We thank the referee for giving constructive suggestions. To address your concerns, we have made the following improvements:

1. In the revised Supplementary Information (highlighted in red, pages 5-6), we have improved the presentation of the concept of magic angle in Fig. S2. We offer a more straightforward and concise description of the system related to the magic angle, making it accessible to readers. In fact, the results in Fig.S2 represent the theoretical analysis forecasting the alteration trend in the effective conductivity. Fig.S2 is just a detailed supplementary explanation that illustrate the oscillation of effective thermal conductivity (it initially decreases and then increases with an increase in the twisted angle). The main discovery is the proposed twisted thermotics in a diffusion system, in which twisting an analog thermal magic angle would result in the thermal conductivity changing and thermal function switching. All those major claims and results have been validated in the main text.
2. In the revised manuscript (highlighted in red, page 5), we provide a more detailed and comprehensible explanation of the magic angle concept, elucidating how it relates to the theory presented.

In all, we are grateful for the referee's critical comments. We have taken the suggestions into account and substantially improved the contents and expressions.

REVIEWERS' COMMENTS

Reviewer #2 (Remarks to the Author):

The authors have addressed well all my comments, the manuscript can be accepted for publication in the present form.

Reviewer #2 (Remarks to the Author):

The authors have addressed well all my comments, the manuscript can be accepted for publication in the present form.

Response: Thanks a lot for the careful review and constructive suggestion